# Deep learning and genome-wide association meta-analyses of bone marrow adiposity in the UK Biobank

Wei Xu [1,2,14], Ines Mesa-Eguiagaray[1,14], David M. Morris[2,3], Chengjia Wang[3,4], Calum D. Gray [3], Samuel Sjöström[2], Giorgos Papanastasiou [3,5], Sammy Badr [6], Julien Paccou[6], Xue Li [7], Paul R. H. J. Timmers[8], Maria Timofeeva [8,9], Susan M. Farrington [10,11], Malcolm G. Dunlop [10,11], Scott I. Semple[2,3], Tom MacGillivray [12], Evropi Theodoratou [1,13] ✉ & William P. Cawthorn [2] ✉

Bone marrow adipose tissue is a distinct adipose subtype comprising more than 10% of fat mass in healthy humans. However, the functions and pathophysiological correlates of this tissue are unclear, and its genetic determinants remain unknown. Here, we use deep learning to measure bone marrow adiposity in the femoral head, total hip, femoral diaphysis, and spine from MRI scans of approximately 47,000 UK Biobank participants, including over 41,000 white and over 6300 non-white participants. We then establish the heritability and genome-wide significant associations for bone marrow adiposity at each site. Our meta-GWAS in the white population finds 67, 147, 134, and 174 independent significant single nucleotide polymorphisms, which map to 54, 90, 43, and 100 genes for the femoral head, total hip, femoral diaphysis, and spine, respectively. Transcriptome-wide association studies, colocalization analyses, and sex-stratified meta-GWASes in the white participants further resolve functional and sex-specific genes associated with bone marrow adiposity at each site. Finally, we perform a multi-ancestry meta-GWAS to identify genes associated with bone marrow adiposity across the different bone regions and across ancestry groups. Our findings provide insights into BMAT formation and function and provide a basis to study the impact of BMAT on human health and disease.

The bone marrow (BM) is a major site of fat storage in species ranging from fish to mammals[1]. Collectively, adipocytes within the BM form BM adipose tissue (BMAT), which accounts for >70% of BM volume and 10% of total fat mass in lean, healthy adult humans. BMAT further increases in diverse diseases and iatrogenic contexts, including osteoporosis, obesity and type 2 diabetes, radiotherapy, and glucocorticoid treatment[2]. Intriguingly, BMAT also increases in conditions of energy deficit, such as anorexia nervosa or caloric restriction[2–6].

BMAT's molecular and functional properties are site-specific, differing between the axial and appendicular skeleton[7,8]. Despite the potential physiological and clinical significance of BMAT, its study has been limited, especially in comparison to white and brown adipose tissues[2]; hence, BMAT formation and function remains relatively poorly understood.

Magnetic resonance imaging (MRI), including MRI with spectroscopy (MRS), remains the gold standard method for non-invasive

measurement of BM adiposity in humans[9,10]. Using MRS and chemical shift-encoding based water-fat separation methods, MRI allows quantification of the BM fat fraction (BMFF). BMFF has been measured in numerous smaller-scale human cohort studies, revealing some insights into BMAT's association with human skeletal and metabolic health[11,12]. For example, using MRS at the lumbar spine, higher BM adiposity is associated with morphometric vertebral fractures and lower bone mineral density (BMD) in osteoporotic and non-osteoporotic subjects[11–13]. However, these studies have never analyzed more than 729 participants[14], limiting the ability to detect other associations. Measuring BMFF at a population-scale therefore has huge potential to reveal fundamental new knowledge of BMAT biology.

Such large-scale BMFF analysis is now possible using the UK Biobank (UKBB). In what is the world's largest health imaging study, 100,000 UKBB participants are undergoing MRI of the brain, heart and whole body, as well as dual-energy X-ray absorptiometry (DXA) to measure BMD[15]. Using these MRI data, we have already established deep learning to efficiently measure BMFF of the spine, femoral head, total hip, and femoral diaphysis[14]. These four sites cover the axial and appendicular skeleton and include major sites of fracture burden, ensuring that the BMFF measurements allow detection of site-specific and clinically relevant BMAT characteristics. We have since applied our deep learning models to complete these multi-site BMFF measurements in over 45,000 individuals and conducted meta-analysis of genome-wide association studies (GWAS) to identify the genes associated with altered BMFF at each site. The findings of these studies, which are the largest analysis to date of BM adiposity, are reported herein.

## Results

### Deep learning analysis of BMFF in the femoral head, total hip, femoral diaphysis, and spine of over 45,000 participants

We used our validated deep learning models to measure the BMFF of the femoral head, total hip, femoral diaphysis, and spine of participants in the UKBB multi-modal imaging study[14,15] (Fig. 1A). The BMFF measurements were conducted in two batches based on the availability of MRI data released by UKBB; together, these two batches comprised 50,226 participants. After sample quality control, for GWAS we retained 38,581 white and 5933 non-white participants for femoral head, 38,394 white and 6047 non-white participants for total hip, 37,513 white and 5844 non-white participants for femoral diaphysis, and 41,204 white and 6367 non-white participants for spine (Table 1; Supplementary Data 1–6). Together, these data represent 48,608 UKBB participants (Fig. 1A). The baseline characteristics of the study samples for each bone region are summarized in Table 1.

Before conducting a meta-GWAS for each bone region, we first assessed if our deep-learning-derived BMFF measurements show expected anatomical characteristics and physiological associations (Fig. 1B). Across the full cohort and for both sexes, spinal BMFF was significantly lower than the BMFF of each femoral region (Fig. 2A; Table 1), consistent with previous studies[12,14,16]. BMFF also differed significantly between each of the three femoral regions, being highest in the femoral head and lowest in the diaphysis (Fig. 2A; Table 1). Nevertheless, there were significant correlations between BMFF at each site: these were strongest between the three femoral regions but weaker between the spine and each femoral site (Fig. 2B). Previous reports identified age-dependent sex differences in spinal BMFF[14,17]. Consistent with this, we found that spinal BMFF was higher in males than females aged 40-49 but higher in females than in males for all other age groups (Fig. 2C; Supplementary Data 8). In contrast, BMFF for each femoral region was greater in males than females, irrespective of age (Fig. 2D-F; Supplementary Data 8).

Ethnicity influences body fat distribution[18] and there may be ethnic differences in BM adiposity[2]. To test this, we compared BMFF at each site between white and non-white participants (Fig. 1B), as classified by UKBB based on genetic ethnic grouping. For both sexes, white participants had greater spine BMFF and lower diaphysis BMFF than non-white participants. In contrast, BMFF of the femoral head and total hip did not differ between white and non-white participants (Supplementary Fig. 1, Supplementary Data 7). These ethnicity-related differences, which were controlled for age and BMI, generally persisted after further controlling for BMD at the respective skeletal sites (spine and femoral shaft), although this slightly diminished the diaphysis difference in males (Supplementary Data 7). To better understand these differences, we further analyzed Asian, Black, and non-white mixed-ethnicity sub-groups, based on self-reported ethnic identities for each participant. For each sex, spine BMFF remained significantly higher in white participants than in Black or non-white mixed-ethnicity participants but did not differ between white and Asian participants. Spine BMFF was also lower in Black participants than in those of Asian or non-white mixed ethnicity (Supplementary Fig. 1A). A similar pattern occurred for femoral head BMFF in females, whereas in males femoral head BMFF differed only between Asian vs white or non-white mixed-ethnicity participants, in each case being lower in those of Asian ethnicity. However, significant differences between Black and white or Asian and white participants emerged after adjusting for BMD at each site (Supplementary Data 7).

We further investigated associations between BMFF and other key physiological characteristics (Fig. 1B). At each site, BMFF was positively associated with age (Fig. 2C–F; Supplementary Data 8,9) and inversely associated with BMD (Fig. 3A–D); the latter was strongest for the spine and diaphysis. These robust associations are consistent with numerous previous studies[2], demonstrating the reliability of our BMFF measurements. Because some studies report increased BMFF in obesity[2,19], we further assessed if BMFF at each site is associated with BMI or peripheral adiposity. BMFF at each femoral site was negatively associated with BMI and waist-hip ratio (WHR) [Supplementary Data 9], regardless of controlling for BMD. The associations between femoral BMFF and other adiposity traits were more complex. MRI-measured visceral adipose tissue volume (VATi) was positively associated with femoral head BMFF, especially in females, but negatively associated with total hip or diaphysis BMFF. In contrast, after adjusting for BMD at each site, MRI-measured abdominal subcutaneous adipose tissue volume (ASATi) was negatively associated with femoral head and diaphysis BMFF but positively associated with total hip BMFF; for the femoral head and total hip, these associations with ASATi were stronger in males than in females (Supplementary Data 9). In females, femoral head BMFF was also positively associated with DXA-measured total body fat % and % fat mass in the trunk, android and gynoid regions, but in males these associations were either negative or insignificant, especially after controlling for BMD (Supplementary Data 9). In contrast, leg fat % was positively associated with femoral head BMFF in both sexes and this was diminished after controlling for BMD. For total hip BMFF, in both sexes there were positive associations with total body fat % and % fat in the gynoid, leg, trunk, or android regions; the latter two remained significant only in females after controlling for BMD. Diaphysis BMFF was negatively associated with total or regional % fat mass, particularly in females, whereas in males it was positively associated with gynoid or leg fat %. The latter association was no longer significant after controlling for BMD (Supplementary Data 9). At the spine, BMFF was positively associated with BMI in males but inversely associated with BMI in both sexes or in females only; after controlling for BMD, these associations became strongly positive for males, females, and both sexes combined. Spinal BMFF was also positively associated with WHR, VATi, ASATi, and total or regional fat mass %, with each of these associations persisting after controlling for spine BMD. Thus, unlike for the femoral sites, spinal BMFF is positively associated with BMI and all other indices of peripheral adiposity. This confirms and greatly extends previous reports of

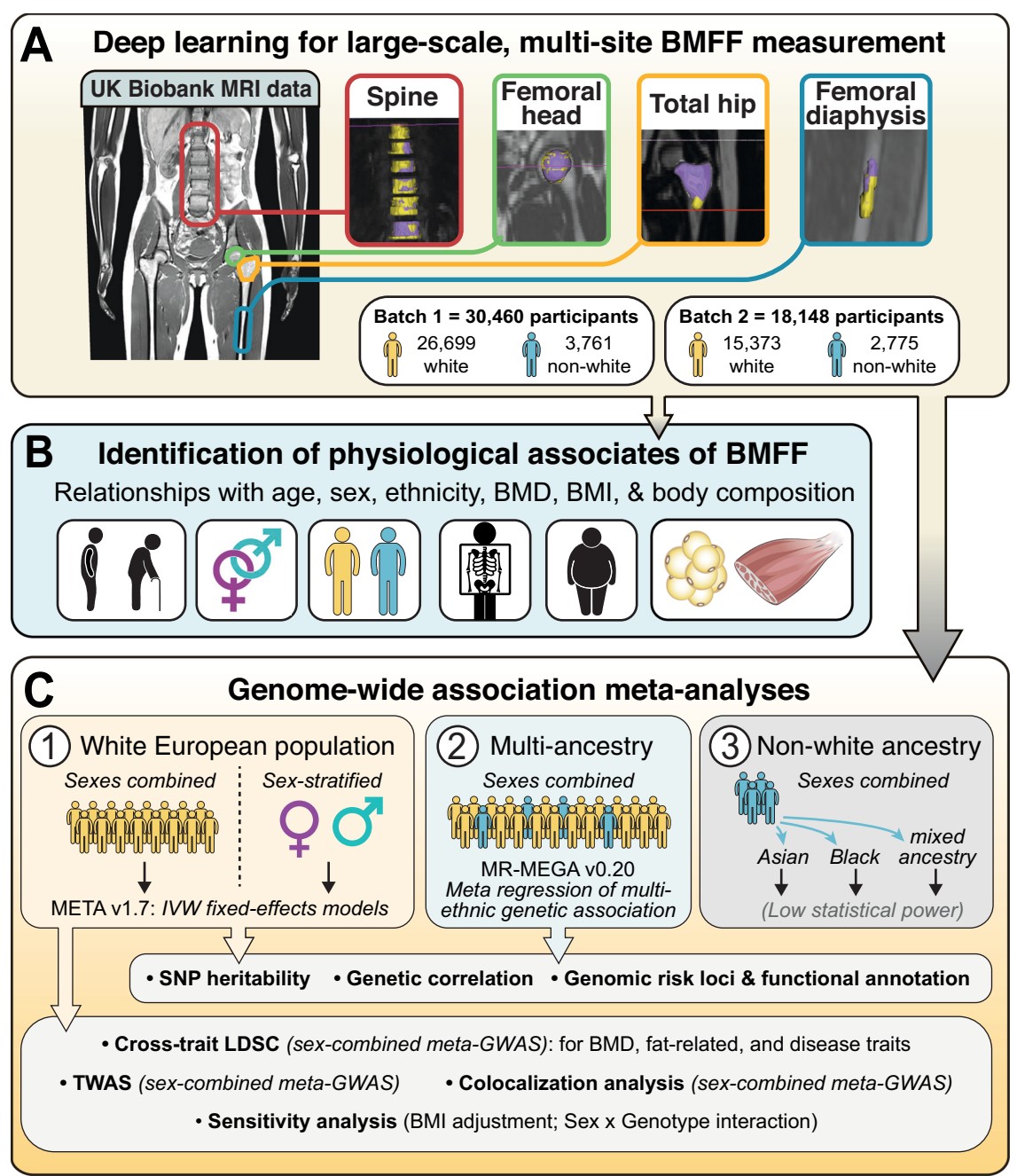

**Fig. 1 | Study design. A** Deep learning was used to segment the spine, femoral head, total hip, and femoral diaphysis from MRI scans of participants in the UKBB imaging study, allowing BMFF measurements at each site. **B** BMFF descriptive and association analysis. **C** GWAS meta-analysis of BMFF. Figure generated using Adobe Illustrator, with some graphics adapted from svgrepo.com under an open license.

increased spine BMFF in obesity and inverse associations between femoral BMFF and ASAT.

## Meta-GWAS in the white population

We conducted meta-GWAS analyses (combining the two batches of BMFF measurements) in the population of unrelated white UKBB participants to identify genetic variants associated with the BMFF of each bone region (Supplementary Data 3-4). A total of 11,353,112 genetic variants passed quality control [imputation quality score (INFO) > 0.4 and minor allele frequency (MAF) > 0.005]. We used inverse-variance-weighted (IVW) fixed effects models to meta-analyze the two GWASes after adjusting for age at imaging visit, sex, BMI at imaging visit, genotyping batch, and population structure of the first

40 principal components (PCs 1-40). GWAS sample quality control details are described in Supplementary Data 1, 2 and the basic characteristics of the white unrelated GWAS samples are summarized in Supplementary Data 3, 4. The SNP-based heritability ($h^2_{SNP}$) of GWAS for each batch, estimated by LDSC, $\lambda_{GC}$, and genetic correlation ($r_g$) results are presented in Supplementary Data 10 and Supplementary Figs. 2–9.

In the femoral head meta-GWAS, we identified 67 independent SNPs ($r^2 < 0.6$) and 23 lead SNPs ($r^2 < 0.1$), residing in 18 genomic risk loci, with $I^2 < 65\%$, that reached genome-wide significance ($P < 5 \times 10^{-8}$) [Fig. 4; Supplementary Data 11, 12]. The $h^2_{SNP}$ of BMFF estimated by LDSC from meta-analysis results was 19.99% (SE = 2.27%), and the LDSC intercept approximated 1 (intercept_meta=1.008, SE = 0.009),

**Table 1 | Summary of the characteristics of the study samples for the four bone regions**

|  | Femoral head (n = 44,514) | Total hip (n = 44,441) | Diaphysis (n = 43,357) | Spine (n = 47,571) |
|---|---|---|---|---|
| Age at imaging (years)* | 65.61 (59.15–71.15) | 65.78 (59.33–71.27) | 65.54 (59.12–71.08) | 65.59 (59.15–71.15) |
| Sex |  |  |  |  |
| Male | 20767 (46.65%) | 20205 (45.46%) | 19516 (45.01%) | 23081 (48.52%) |
| Female | 23747 (53.35%) | 24236 (54.54%) | 23841 (54.99%) | 24490 (51.48%) |
| Ancestry |  |  |  |  |
| White | 38581 (86.67%) | 38394 (86.39%) | 37513 (86.52%) | 41204 (86.62%) |
| Non-white | 5933 (13.33%) | 6047 (13.61%) | 5844 (13.48%) | 6367 (13.38%) |
| BMI at imaging (kg/m²)* | 26.02 (23.61–28.92) | 25.84 (23.43-28.76) | 26.00 (23.56–28.92) | 25.96 (23.54–28.90) |
| BMFF measurement |  |  |  |  |
| BMFF (%)* | 91.81 (90.52–92.69) | 91.45 (89.06–92.95) | 82.68 (79.43–85.46) | 51.78 (45.96–56.96) |
| Size of segmented region (voxels)* | 737.00 (598.00–929.00) | 1034.00 (790.00–1334.00) | 96.00 (80.00–108.00) | 2103.00 (1810.00–2404.00) |

*Values are presented as median (interquartile range). Numbers of participants in each sex and ancestry category are shown as absolute number, with % of total for each bone region indicated in parentheses.

suggesting that the inflation ($\lambda_{GC}$_meta=1.114) in the meta-analysis was consistent with polygenicity. The directions of effect sizes for the reported associations were concordant in the meta-GWAS and the GWASes for the two individual batches. FUMA was used to map the candidate genetic variants implicated in the femoral head meta-GWAS based on positional mapping, which identified a total of 54 mapped genes (Fig. 4; Supplementary Data 13).

In the total hip meta-GWAS, we identified 147 independent SNPs ($r^2 < 0.6$) and 48 lead SNPs ($r^2 < 0.1$), residing in 29 genomic risk loci, with $I^2 < 65\%$, that were genome-wide significant ($P < 5 \times 10^{-8}$) [Fig. 4; Supplementary Data 11-12]. LDSC results suggested that the signals were primarily driven by polygenicity with $h^2_{SNP} = 27.37\%$ (SE = 2.28%), intercept_meta=1.019 (SE = 0.008), and $\lambda_{GC}$_meta = 1.159. The GWAS results of each batch showed high consistency with the meta-GWAS results. FUMA positional mapping identified 90 genes in the total hip meta-GWAS (Fig. 4; Supplementary Data 13).

In the femoral diaphysis meta-GWAS, we identified 134 independent SNPs ($r^2 < 0.6$) and 37 lead SNPs ($r^2 < 0.1$), residing in 18 genomic risk loci, with $I^2 < 65\%$, that reached genome-wide significance ($P < 5 \times 10^{-8}$) [Fig. 4; Supplementary Data 11, 12]. Genomic inflation was moderate ($\lambda_{GC}$_meta=1.146) and consistent with polygenicity ($h^2_{SNP} = 27.52\%$, SE = 2.48%; intercept_meta=1.009, SE = 0.009). The directions of effect sizes for the reported associations remained consistent across the two batches and the meta-GWAS. We found 43 mapped genes in the femoral diaphysis meta-GWAS (Fig. 4; Supplementary Data 13).

In the spine meta-GWAS, we identified 174 independent SNPs ($r^2 < 0.6$) and 51 lead SNPs ($r^2 < 0.1$), residing in 38 genomic risk loci, with $I^2 < 65\%$ that were genome-wide significant ($P < 5 \times 10^{-8}$) [Fig. 4; Supplementary Data 11, 12]. The $h^2_{SNP}$ of BMFF estimated by LDSC was 24.98% (SE = 2.65%). The LDSC intercept of 1.014 (SE = 0.011) was close to 1, suggesting that the observed genomic inflation ($\lambda_{GC}$_meta=1.146) was primarily due to polygenicity rather than population stratification. We found high consistency in the directions of effect sizes for the reported associations in the two batches and the meta-GWAS. FUMA positional mapping found 100 genes for the spine meta-GWAS (Fig. 4; Supplementary Data 13).

In addition, to assess the impact of BMI adjustment on the BMFF trait associations, we compared the meta-GWAS results with and without BMI adjustment for the four bone regions. Our analysis did not find any distinct discrepancies in the results (Supplementary Data 14–18). The BMFF association beta values remained in the same direction, with a median beta difference of around $1.6 \times 10^{-5}$ (Supplementary Data 15, 17).

Based on gene mapping from our meta-GWAS with BMI adjustment, one gene (*TIMP4*) was identified in all four bone regions, 10 genes in three bone regions (*LEPR, LEPROT, PPARG, TERT, CCDC170, ESR1, COLEC10, TNFRSF11B, TNFSF11, AKAP11*), and 34 genes in two bone regions (Table 2; Fig. 5). Many of the genetic associations were unique to each region, including 37 unique genes for the femoral head, 48 for the total hip, 25 for the diaphysis, and 73 for the spine (Fig. 5; Supplementary Data 13).

Furthermore, we tested more-stringent thresholds for functional annotation of the Meta-GWAS. Supplementary Data 19, 20 present results based on a genome-wide significant $P$-value < $5 \times 10^{-8}$, with LD thresholds of $r^2 < 0.3$ for independent significant SNPs and $r^2 < 0.1$ for lead SNPs. When comparing the independent significant SNPs identified using different LD thresholds ($r^2 < 0.3$ versus $r^2 < 0.6$) at a significance level of $P$-value < $5 \times 10^{-8}$, we found that the more-stringent threshold ($r^2 < 0.3$) resulted in approximately half the number of independent significant SNPs [Femoral head: $n = 34$; Total hip: $n = 72$; Diaphysis: $n = 65$; Spine: $n = 80$] compared to the less-stringent threshold across all bone regions [Femoral head: $n = 67$; Total hip: $n = 147$; Diaphysis: $n = 134$; Spine: $n = 174$] (Supplementary Data 20). Despite this reduction, all SNPs (lead and independent significant SNPs) identified under the more-stringent threshold were included among those identified using the less-stringent threshold (Supplementary Data 23), reinforcing the reliability of our findings. In addition, the same LD $r^2$ thresholds were used in Supplementary Data 21, 22, but with a genome-wide significant $P$-value < $1 \times 10^{-8}$. Overall, there was a high degree of overlap of SNPs identified across different LD and P-value thresholds: although the more-stringent threshold ($r^2 < 0.3$; P-value < $1 \times 10^{-8}$) reduced the number of identified independent SNPs, these SNPs were largely consistent with those identified with the less-stringent threshold ($r^2 < 0.6$; P-value < $5 \times 10^{-8}$) [Supplementary Data 21–23]. Thus, the more-stringent thresholds do not substantially alter the main findings, reinforcing the reliability of our results obtained using LD $r^2 < 0.6$ and P-value < $5 \times 10^{-8}$.

## Gene to function for meta-GWAS in the white population

To understand putative biological mechanisms and functional roles of BMFF-associated variants, we first performed MAGMA gene-set analysis, tissue expression analysis, and cell-type-specific gene expression analysis[20,21]. In the MAGMA gene-set results (Fig. 6), two gene-sets were associated with femoral head BMFF: breast cancer 20q11 amplicon ($P = 2.66 \times 10^{-24}$) and breast cancer 20q13 amplification up ($P = 6.68 \times 10^{-6}$). Two gene-sets were associated with total hip

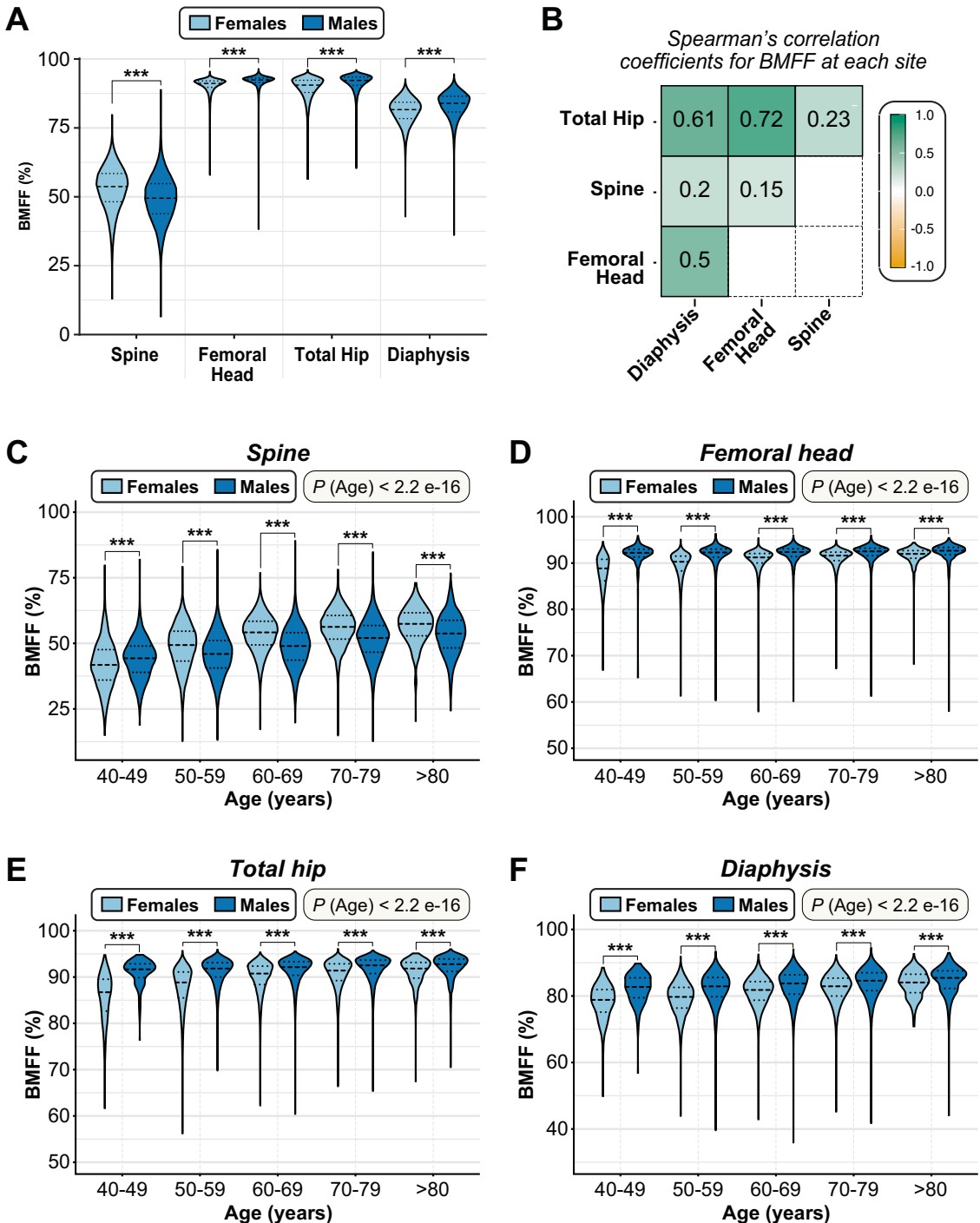

**Fig. 2 | BMFF comparison between age and sex in four bone regions. A** Summary of BMFF (%) at each site, shown separately for males and females. Data are presented as violin plots, with the median shown as a dashed horizontal line and the 25% and 75% as dotted horizontal lines, of the following numbers of participants: spine – female, 24,490; spine – male, 23,081; femoral head – female, 23,747; femoral head – male, 20,767; total hip – female, 24,236; total hip – male, 20,205; diaphysis – female, 23,841; diaphysis – male, 19,516. **B** Correlation matrix showing Spearman's rank correlation coefficients between BMFF at each skeletal site in participants where BMFF was measured ($n = 39,632$). Higher values indicate greater similarity in BMFF between sites. **C–F** BMFF at the spine (**C**, $n = 47,571$), femoral head (**D**, $n = 44,514$), total hip (**E**, $n = 44,441$), and femoral diaphysis (**F**, $n = 43,357$) across each decade of age for males and females, presented as for **A**. For **A** and **C–F**, significant sex differences within each site (**A**) or decade of age (**C–F**), and significant site differences within each sex (**A**), were assessed using multivariate ANOVA to compare rank-normalized BMFF values, controlling for BMI and age at imaging (**A**) or BMI only (**C-F**). In (**A**), BMFF significantly differed between each site (for each pairwise comparison $P < 2.2e-16$). Sex differences in **A** and **C–F** are indicated by *** ($P < 0.001$). See Supplementary Data 8 for further details.

BMFF: breast cancer 7q21 q22 amplicon ($P = 5.06 \times 10^{-10}$) and osteoclast signaling ($P = 3.82 \times 10^{-5}$). Eleven gene-sets were associated with femoral diaphysis BMFF, with the top associations being breast cancer luminal A up ($P = 1.10 \times 10^{-6}$), osteoclast signaling ($P = 6.31 \times 10^{-6}$), clock-controlled autophagy in bone metabolism ($P = 3.28 \times 10^{-5}$), and type 1 collagen synthesis in the context of osteogenesis imperfecta ($P = 4.94 \times 10^{-5}$). Six gene-sets were associated with spine BMFF: the top associations were breast cancer

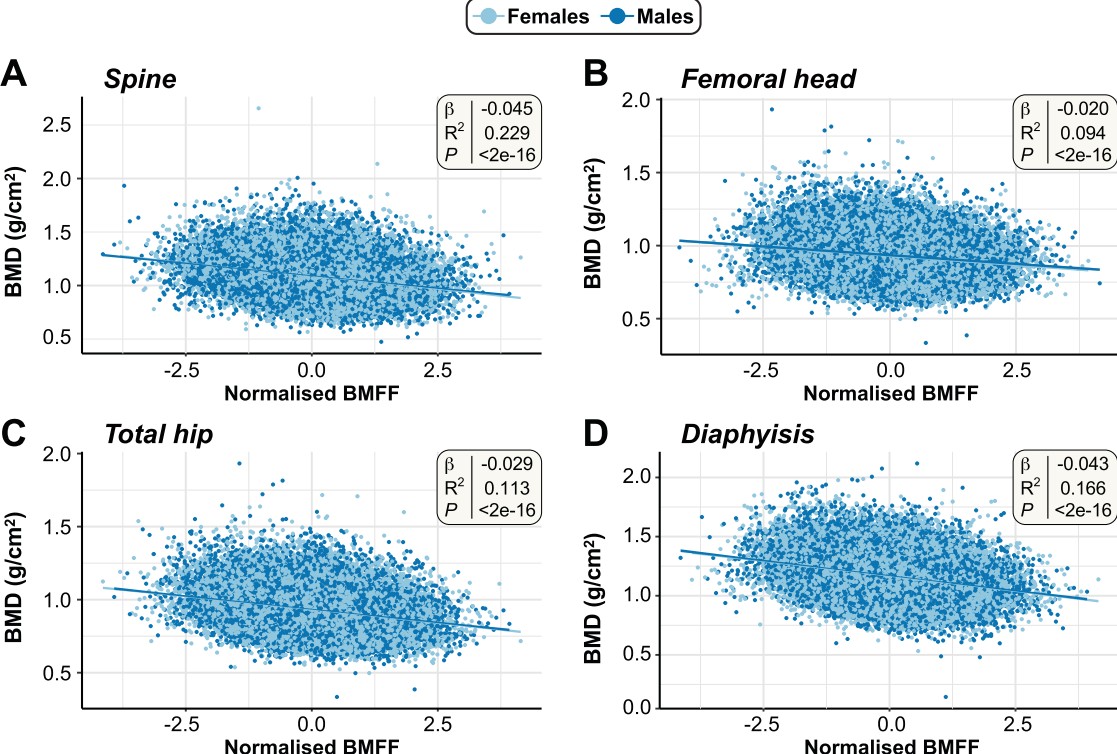

**Fig. 3 | Associations between BMFF and bone mineral density at each site.**
Linear regression for rank-normalized BMFF vs BMD (g/cm²) at each skeletal site, as follows: **A** Spine BMFF vs spine BMD ($n = 40,485$); **B** Femoral head BMFF vs femoral neck (left) BMD ($n = 37,967$); **C** Total hip BMFF vs femoral trochanter (left) BMD ($n = 38,007$); **D** Femoral diaphysis BMFF vs femoral shaft (left) BMD ($n = 37,015$). The smaller sample sizes for the linear regressions between BMFF and BMD at the respective skeletal sites compared to the analyses in Fig. 2 are the result of not all participants (for whom BMFF was measured) having complete BMD measurements from DXA-scans. Linear regression summary statistics (Beta coefficients [β], R-squared [R²] and p-values [p]) for the multivariate linear regression models, adjusted for Sex, Age and BMI, are shown in the top-right corner of each plot.

1q21 amplicon ($P = 7.08 \times 10^{-10}$) and large intestine adult OLFM4 high stem cell ($P = 3.11 \times 10^{-6}$) (Fig. 6). MAGMA tissue expression analysis revealed the gene expression profiles of the BMFF-associated genes in 54 tissue types. We did not find any significant tissue associations after Bonferroni correction for any of the BMFF regions (Supplementary Data 24). MAGMA gene-property analysis, focusing on BM cell types, demonstrated that the strongest associations were found in BM mesenchymal fibroblasts ($P = 8.68 \times 10^{-5}$) for femoral head BMFF; BM c-kit macrophages (C1qc high; $P = 0.011$) for total hip BMFF; BM c-kit eosinophil progenitor cells ($P = 0.009$) for femoral diaphysis BMFF; and BM mesenchymal endothelial cells (Ly6c1 high; $P = 0.029$) for spine BMFF (Supplementary Data 25).

**Transcriptome-wide association studies (TWAS)**
We conducted TWAS to identify risk genes whose genetically regulated expression levels are associated with BMFF; TWAS Hub (http://twas-hub.org) confirmed that no previous TWASes have been done for BMFF. Because GTEx does not yet include BM or bone tissues, our TWAS was based on gene expression prediction models generated from subcutaneous adipose tissue, visceral-omentum adipose tissue and skeletal muscle (GTEx v8), i.e., mesodermal tissues related to adipose and musculoskeletal biology.

Across all three of these tissues (subcutaneous adipose tissue, visceral-omentum adipose tissue and skeletal muscle), TWAS identified 31, 49, 32, and 64 genes that, after Bonferroni correction, were significantly associated with BMFF of the femoral head (Supplementary Data 26; Supplementary Fig. 10), total hip (Supplementary Data 27; Supplementary Fig. 11), diaphysis (Supplementary Data 28; Supplementary Fig. 12) and spine

(Supplementary Data 29; Supplementary Fig. 13), respectively. For each BMFF site, the number of significant TWAS genes within each GTEx tissue was as follows: femoral head – subcutaneous adipose ($n = 8$), visceral-omentum adipose ($n = 11$), skeletal muscle ($n = 12$); total hip – subcutaneous adipose ($n = 17$), visceral-omentum adipose ($n = 16$), skeletal muscle ($n = 16$); diaphysis – subcutaneous adipose ($n = 12$), visceral-omentum adipose ($n = 7$), skeletal muscle ($n = 13$); spine – subcutaneous adipose ($n = 26$), visceral-omentum adipose ($n = 19$), skeletal muscle ($n = 19$). These included several genes not identified by our meta-GWAS, including 8 genes for femoral head (*EYA1, TACC3, RP11-179B2.2, NFS1, PHF20, NORAD, RP11-1398P2.1*, and *MMP24-AS1*), 11 for total hip (*ARPC1A, DLX6-AS1, EYA1, IRS1, SERTAD4, RP11-425D17.1, CNPY4, JAZF1-AS1, UCK1, TRIM38*, and *NFS1*), 11 for diaphysis (*DLX6-AS1, CYP19A1, CEBPZ, GPR17, MEGF9, SYN2, SEC11A, FBXW2, TCAP, PSMD5-AS1*, and *EPSTI1*), and 16 for spine BMFF (*GBA, RIMKLBP2, TBC1D8-AS1, RP11-392O17.1, KLRK1, IL18RAP, KLRC3, RP11-219G17.9, TSNAXIP1, MAPK4, THA1P, UBE2L3, AK4, KRT18P17, LIME1*, and *XRCC3*).

Among all of the BMFF-associated genes identified by TWAS, 11 were common to at least two bone regions (Supplementary Data 30). In particular, *UQCC1* was identified for the femoral head [subcutaneous adipose, $Z = -8.811$, $P = 1.24 \times 10^{-18}$; visceral-omentum adipose, $Z = -8.758$, $P = 1.99 \times 10^{-18}$; skeletal muscle $Z = -8.373$, $P = 5.62 \times 10^{-17}$], total hip [subcutaneous adipose, $Z = -4.997$, $P = 5.82 \times 10^{-7}$; visceral-omentum adipose, $Z = -4.920$, $P = 8.67 \times 10^{-7}$; skeletal muscle, $Z = -4.646$, $P = 3.38 \times 10^{-6}$] and diaphysis [subcutaneous adipose, $Z = -4.709$, $P = 2.49 \times 10^{-6}$; visceral-omentum adipose, $Z = -4.732$, $P = 2.23 \times 10^{-6}$].

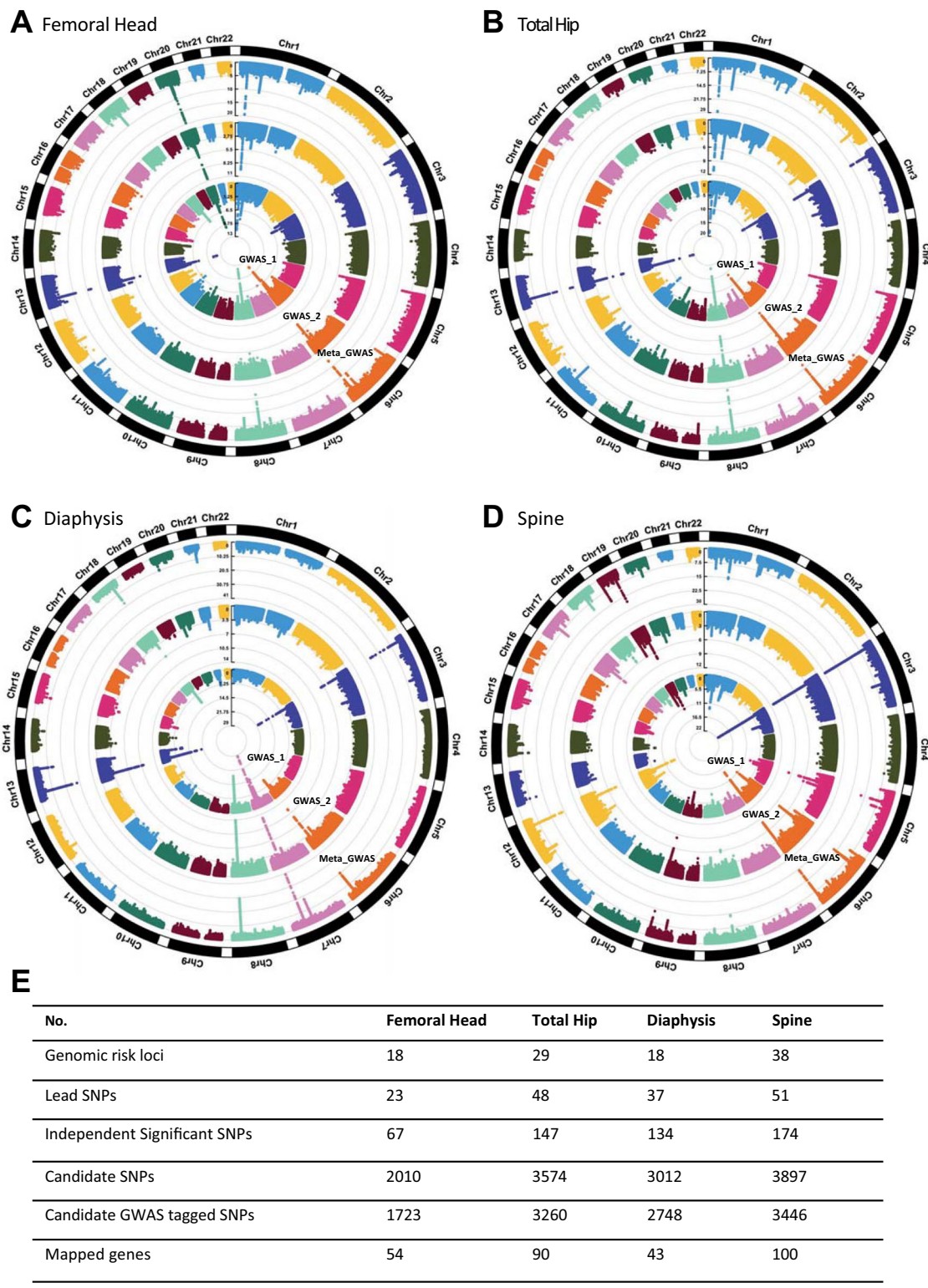

**Fig. 4 | SNP-based associations with BMFF in meta-GWAS for the white unrelated population.** Radial Manhattan plots for results of GWASes for BMFF of the femoral head (**A**), total hip (**B**), diaphysis (**C**) and spine (**D**) in white participants. For each plot, the inner, middle, and outer circles show results from the 1st batch, 2nd batch, and meta-GWAS, as indicated. **E** Summary of the numbers of significant loci and mapped genes for each site. Further details, including standard Manhattan plots and QQ plots for each batch for each site, are shown in Supplementary Figs. 1–8.

| No. | Femoral Head | Total Hip | Diaphysis | Spine |
|---|---|---|---|---|
| Genomic risk loci | 18 | 29 | 18 | 38 |
| Lead SNPs | 23 | 48 | 37 | 51 |
| Independent Significant SNPs | 67 | 147 | 134 | 174 |
| Candidate SNPs | 2010 | 3574 | 3012 | 3897 |
| Candidate GWAS tagged SNPs | 1723 | 3260 | 2748 | 3446 |
| Mapped genes | 54 | 90 | 43 | 100 |

*NT5DC2* also showed significant associations for femoral head [subcutaneous adipose, $Z = -5.190$, $P = 2.10 \times 10^{-7}$; visceral-omentum adipose, $Z = -4.936$, $P = 7.97 \times 10^{-7}$], total hip [subcutaneous adipose, $Z = -4.842$, $P = 1.28 \times 10^{-6}$; visceral-omentum adipose, $Z = -4.815$, $P = 1.47 \times 10^{-6}$] and diaphysis [subcutaneous adipose, $Z = -5.250$, $P = 1.52 \times 10^{-7}$; visceral-omentum adipose, $Z = -4.681$, $P = 2.85 \times 10^{-6}$].

*GSDMA* was common in total hip [subcutaneous adipose, $Z = 5.302$, $P = 1.15 \times 10^{-7}$; visceral-omentum adipose, $Z = 5.266$, $P = 1.40 \times 10^{-7}$; skeletal muscle, $Z = 4.761$, $P = 1.92 \times 10^{-6}$] and spine [subcutaneous

**Table 2 | Mapped genes common to 3 or 4 bone regions in meta-GWAS for the white unrelated population**

| Gene | Bone region | Chr | pLI score | ncRVIS score | Pos Map SNPs | posMap MaxCADD | min GwasP |
|------|-------------|-----|-----------|--------------|--------------|----------------|-----------|
| TIMP4 | femoral head | 3 | 0.0036 | −0.2373 | 15 | 12.03 | 1.10E−07 |
| TIMP4 | total hip | 3 | 0.0036 | −0.2373 | 25 | 13.82 | 2.38E−10 |
| TIMP4 | diaphysis | 3 | 0.0036 | −0.2373 | 14 | 13.82 | 1.05E−11 |
| TIMP4 | spine | 3 | 0.0036 | −0.2373 | 16 | 12.03 | 1.33E−09 |
| LEPR | femoral head | 1 | 0.9998 | 0.9275 | 199 | 12.84 | 1.76E−11 |
| LEPR | total hip | 1 | 0.9998 | 0.9275 | 281 | 18.75 | 1.72E−16 |
| LEPR | spine | 1 | 0.9998 | 0.9275 | 297 | 18.75 | 8.44E−16 |
| LEPROT | femoral head | 1 | 0.0948 | 1.2665 | 11 | 11.01 | 3.73E−08 |
| LEPROT | total hip | 1 | 0.0948 | 1.2665 | 11 | 11.01 | 1.05E−10 |
| LEPROT | spine | 1 | 0.0948 | 1.2665 | 13 | 11.59 | 4.63E−09 |
| PPARG | total hip | 3 | 0.6682 | −0.3317 | 37 | 16.21 | 2.59E−15 |
| PPARG | diaphysis | 3 | 0.6682 | −0.3317 | 132 | 17.82 | 6.77E−28 |
| PPARG | spine | 3 | 0.6682 | −0.3317 | 90 | 16.24 | 1.29E−15 |
| TERT | femoral head | 5 | 0.8662 | NA | 13 | 4.268 | 4.33E−09 |
| TERT | total hip | 5 | 0.8662 | NA | 12 | 3.147 | 2.24E−11 |
| TERT | spine | 5 | 0.8662 | NA | 7 | 3.147 | 2.09E−08 |
| CCDC170 | femoral head | 6 | 1.02E−19 | 1.3807 | 102 | 16.75 | 3.41E−18 |
| CCDC170 | total hip | 6 | 1.02E−19 | 1.3807 | 175 | 19.94 | 1.25E−24 |
| CCDC170 | diaphysis | 6 | 1.02E−19 | 1.3807 | 195 | 19.94 | 1.71E−21 |
| ESR1 | femoral head | 6 | 0.9872 | −1.0569 | 50 | 14.50 | 4.60E−09 |
| ESR1 | total hip | 6 | 0.9872 | −1.0569 | 124 | 18.55 | 1.20E−16 |
| ESR1 | diaphysis | 6 | 0.9872 | −1.0569 | 13 | 9.904 | 4.73E−08 |
| COLEC10 | femoral head | 8 | 0.0985 | 0.2633 | 91 | 17.74 | 2.56E−07 |
| COLEC10 | total hip | 8 | 0.0985 | 0.2633 | 91 | 17.74 | 1.75E−08 |
| COLEC10 | diaphysis | 8 | 0.0985 | 0.2633 | 228 | 21.90 | 4.97E−28 |
| TNFRSF11B | femoral head | 8 | 0.2722 | −0.3525 | 36 | 20.60 | 8.46E−09 |
| TNFRSF11B | total hip | 8 | 0.2722 | −0.3525 | 36 | 20.60 | 2.89E−09 |
| TNFRSF11B | diaphysis | 8 | 0.2722 | −0.3525 | 103 | 20.60 | 6.92E−31 |
| TNFSF11 | femoral head | 13 | 0.1302 | 0.2394 | 1 | 0.735 | 2.14E−08 |
| TNFSF11 | totalhip | 13 | 0.1302 | 0.2394 | 1 | 0.735 | 8.18E−10 |
| TNFSF11 | diaphysis | 13 | 0.1302 | 0.2394 | 126 | 19.36 | 1.33E−25 |
| AKAP11 | femoral head | 13 | 0.8964 | 0.1431 | 44 | 10.87 | 3.67E−06 |
| AKAP11 | totalhip | 13 | 0.8964 | 0.1431 | 35 | 10.87 | 1.15E−09 |
| AKAP11 | diaphysis | 13 | 0.8964 | 0.1431 | 22 | 10.87 | 8.92E−09 |

*Chr*, chromosome; pLI score (from ExAC database) is the probability of being loss-of-function intolerant (the higher the score is, the more intolerant to loss-of-function mutations the gene is); ncRVIS score is the non-coding residual variation intolerance score (the higher the score is, the more intolerant to non-coding variation the gene is); posMapSNPs is the number of SNPs mapped to gene based on positional mapping (after functional filtering if parameters are given); posMapMaxCADD is the maximum CADD score of mapped SNPs by positional mapping; minGwasP is the minimum P-value of mapped SNPs (two-sided). For each gene, the Gene biotype from Ensembl is "protein_coding". Positional mapping was performed based on annotations obtained from ANNOVAR. For space reasons, the following parameters are not shown in this table, but are presented for each gene and region in Supplementary Data 13 (Mapped genes in meta-GWAS in the sample of white unrelated participants): start position; stop position; IndSigSNPs, which shows the rsIDs of the independent significant SNPs that are in LD with the mapped SNPs; and GenomicLocus, which is the index of genomic loci where mapped SNPs are from (multiple loci can be assigned with ":" delimiter). Mapped genes found for one or two bone regions are shown in Supplementary Data 13.

adipose, $Z = 6.896$, $P = 5.34 \times 10^{-12}$; visceral-omentum adipose, $Z = 6.844$, $P = 7.69 \times 10^{-12}$; skeletal muscle, $Z = 6.432$, $P = 1.26 \times 10^{-10}$].

Finally, *CCDC170* presented significant associations for femoral head [subcutaneous adipose, $Z = 4.744$, $P = 2.10 \times 10^{-6}$; visceral-omentum adipose, $Z = 9.458$, $P = 3.14 \times 10^{-21}$], total hip [subcutaneous adipose, $Z = 7.630$, $P = 2.36 \times 10^{-14}$; visceral-omentum adipose, $Z = 13.192$, $P = 9.71 \times 10^{-40}$] and diaphysis [visceral-omentum adipose, $Z = 9.665$, $P = 4.25 \times 10^{-22}$].

## Colocalization

We performed colocalization analysis to further explore whether genetic variants for BMFF are associated with altered gene expression in mesodermal tissues (subcutaneous adipose tissue, visceral adipose tissue, skeletal muscle) and extended this analysis to include lymphoid tissues (spleen, lymphocytes). Using cis-eQTL from GTEx v8, associations were explored within the significant meta-GWAS SNPs. In the femoral head and total hip meta-GWAS, evidence for colocalization was found for the *CCND2* locus in subcutaneous adipose tissue (posterior probability of hypothesis 4 [PPH4] = 1) and skeletal muscle tissue (PPH4 = 1) [Supplementary Data 31]. In the femoral diaphysis meta-GWAS, a locus of high PPH4 (PPH4 = 0.97) was found for *TNFRSF11A* in visceral-omentum adipose tissue, mirroring the TWAS findings for this gene (Supplementary Data 28; Supplementary Fig. 12). In the spine meta-GWAS, four colocalizations were detected, with *RAF1* and *CYP19A1* in skeletal muscle tissue (PPH4 = 1), *EEFSEC* in visceral-omentum adipose tissue (PPH4 = 0.91) and *STXBP6* in spleen tissue (PPH4 = 1) [Supplementary Data 31]. Notably, differential expression of *CYP19A1, EEFSEC,* and *STXBP6* was also identified through TWAS of the

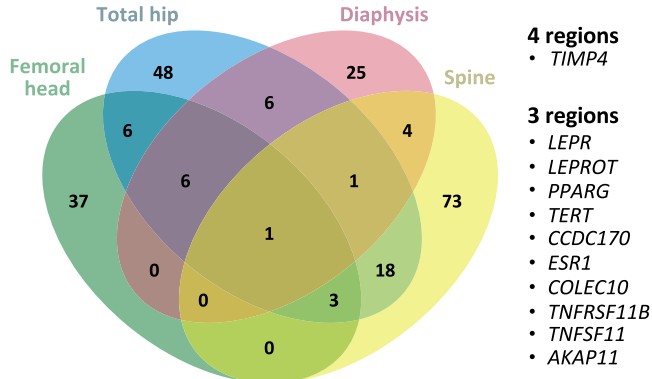

**A** Comparison of overlap mapped genes in four bone regions

**4 regions**
- *TIMP4*

**3 regions**
- *LEPR*
- *LEPROT*
- *PPARG*
- *TERT*
- *CCDC170*
- *ESR1*
- *COLEC10*
- *TNFRSF11B*
- *TNFSF11*
- *AKAP11*

**B**

| Number of overlapping regions | 4 regions | 3 regions | 2 regions |
|---|---|---|---|
| Lead SNPs | 0 | 0 | 10 |
| Independent significant SNPs | 0 | 5 | 36 |
| Mapped genes | 1 | 10 | 34 |

**Fig. 5 | Overlap of mapped genes associated with each BMFF region for meta-GWAS in the white unrelated population. A** Venn diagram showing mapped genes shared between regions or unique to each region. The list beside the diagram shows the gene names shared for three or all four regions. **B** Table showing lead SNPs, independent significant SNPs, and mapped genes that are shared for two or more regions.

loci associated with spine BMFF (Supplementary Data 29; Supplementary Fig. 13).

Together, these TWAS and colocalization results identified risk genes whose genetically regulated expression levels in adipose tissues, skeletal muscle, and/or lymphoid tissues were associated with altered BMFF, thereby extending our meta-GWAS findings.

**Comparison with previous GWAS findings**

To determine if BMFF has a distinct genetic architecture, we used FUMA[20] to identify SNPs and mapped genes with previously reported phenotypic associations (in published GWAS listed in the NHGRI-EBI catalog) that overlapped with the genomic risk loci identified in our meta-GWAS for the white population (Supplementary Data 32–35). In the femoral head, 41 (61.19%) significant SNPs were reported in previous GWASes for traits such as BMD, femoral neck size, hip bone size, osteoporosis, osteoarthritis, type 2 diabetes, and breast cancer. In the total hip, 83 (56.46%) significant SNPs were reported in previous GWASes, including traits such as BMD, hip circumference, hip bone size, osteoporosis, osteoarthritis, cholesterol levels, type 2 diabetes, breast cancer, stroke, and hypertension. In the femoral diaphysis, 62 (46.27%) significant SNPs were reported in previous GWASes for traits such as BMD, osteoporosis, knee osteoarthritis, fractures, type 2 diabetes, coronary artery disease, and cardiovascular disease. In the spine, 103 (59.20%) significant SNPs were reported in previous GWASes for traits such as BMD, osteoporosis, fractures, type 2 diabetes, coronary artery disease, cardiovascular disease, hypertension, ischemic stroke, and chronic lymphocytic leukemia.

We further used cross-trait LDSC to estimate the overall genetic correlation between BMFF and traits related to bone biology (BMD, osteoarthritis), peripheral adiposity (BMI, WHR), cardiometabolic diseases (type 2 diabetes, coronary artery disease, stroke, hypertension), and breast cancer (Supplementary Data 36). These LDSC results showed significant negative genetic correlation between BMFF and BMD in all four bone regions [femoral head: $r_g = -0.260$, $P = 7.61 \times 10^{-7}$; total hip: $r_g = -0.256$, $P = 5.37 \times 10^{-7}$; diaphysis: $r_g = -0.159$, $P = 5 \times 10^{-4}$; spine: $r_g = -0.130$, $P = 1.08 \times 10^{-3}$], consistent with the robust inverse associations between BMFF and BMD at each site (Fig. 3). For BMI, the LDSC regression intercepts showed significant negative correlation for the femoral head ($r_g = -0.252$, $P = 1.75 \times 10^{-17}$), total hip ($r_g = -0.263$, $P = 6.36 \times 10^{-22}$) and, less strongly, the diaphysis ($r_g = -0.078$, $P = 1.90 \times 10^{-3}$), but there was no correlation between BMI and spine BMFF ($r_g = -0.028$, $P = 2.56 \times 10^{-1}$) [based on meta-GWAS without BMI adjustment]. Suggestive genetic correlation was also observed for BMFF and WHR or WHR$_{adj}$BMI, with negative correlations occurring for BMFF at each femoral site but positive correlations for spine BMFF (Supplementary Data 36). The cross-trait LDSC findings showed no significant genetic correlation between BMFF and the other disease traits. Moreover, despite reaching statistical significance, the genetic correlation $r_g$ values between BMFF and BMD, BMI, WHR, or WHR$_{adj}$BMI were moderate (ranging from -0.263 to 0.154), indicating only a partial degree of shared genetic signal overlap. Together, the findings suggest that the genetic determinants of BMFF are distinct from those for traits relating to peripheral adiposity, bone biology, breast cancer, and cardiometabolic diseases.

**Sex sensitivity analysis for meta-GWAS in the white population**

To establish if any BMFF-associated genetic variants are sex-specific, we conducted GWAS meta-analyses, stratified by sex in the unrelated white population, for the four bone regions (Supplementary Data 37, 38).

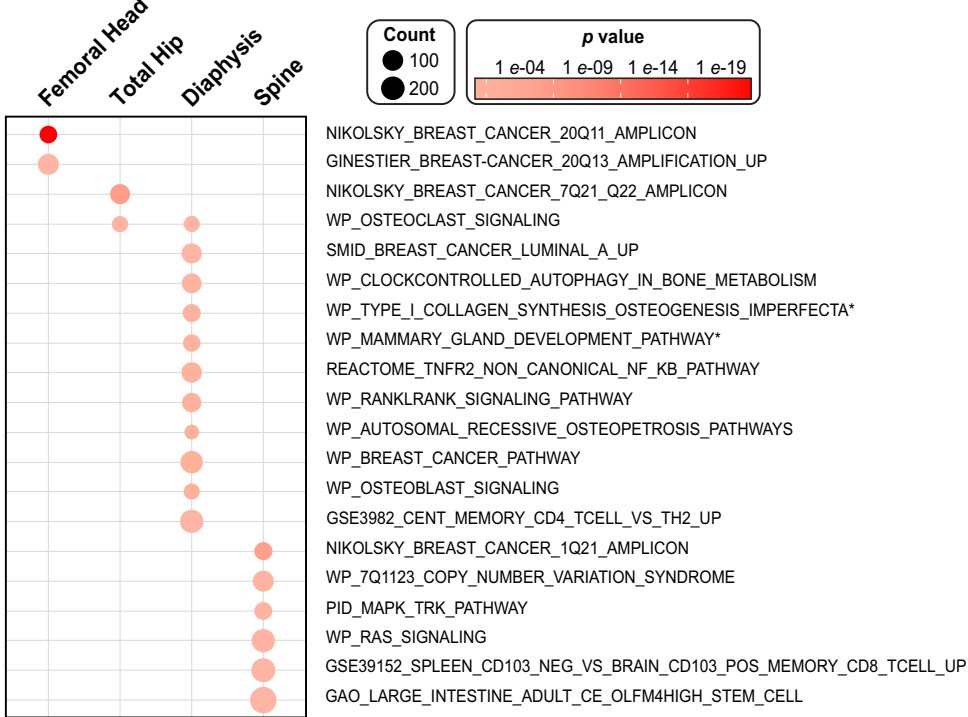

**Fig. 6 | MAGMA Gene-set analysis for Meta-GWAS in the white unrelated population.** Bubble plot for gene sets significantly enriched among the BMFF-associated genes identified by meta-GWAS in the white participants (males and females combined). In MAGMA gene-set analysis (two-sided), each gene set (or GO term) was tested for enrichment. Significance is based on Bonferroni-corrected *p* values, as indicated by the color of each bubble. The size of each bubble represents the number of genes in each gene set. To streamline the presentation, the names of two gene sets (indicated by *) have been abridged to fit within the figure; these gene set's full names are 'WP_TYPE_I_COLLAGEN_SYNTHESIS_IN_THE_CONTEXT_OF_OS-TEOGENESIS_IMPERFECTA' and 'WP_MAMMARY_GLAND_DEVELOPMENT_PATHWAY_PREGNANCY_AND_LACTATION_STAGE_3_OF_4'.

In males, 6, 22, 32 and 31 independent significant SNPs, which mapped to 12, 11, 13 and 24 genes, were associated with BMFF in the femoral head, total hip, diaphysis and spine, respectively. In females, 29, 53, 47 and 46 independent significant SNPs, which mapped to 8, 15, 12, and 31 genes, were found to be associated with BMFF in these respective regions. Full details of sex-specific meta-GWAS, including LDSC results, are presented in Supplementary Data 39–42, Supplementary Figs. 14–21, and Supplementary Information (Note 1).

In addition, to assess the impact of BMI adjustment on the BMFF trait associations, we compared the meta-GWAS results with and without BMI adjustment in males and females. We did not find any distinct discrepancies in the comparisons (Supplementary Data 43–47). The BMFF association beta values remained in the same direction, with median beta differences of around $1.4 \times 10^{-4}$ and $3.4 \times 10^{-4}$ in male and females, respectively (Supplementary Data 44, 46).

Based on gene mapping for the sex-stratified meta-GWAS with BMI adjustment, in males, we identified five genes (*CCDC170, SHFM1, C7of76, RP11-1102P16.1, AKAP11*) that were associated with BMFF in two bone regions. In females, we identified three genes (*LEPR, PPARG, CCDC170*) that were associated with three bone regions and four genes (*TIMP4, RP11-1102P16.1, TNFRSF11B, COLEC10*) with two bone regions (Supplementary Data 42). Notably, we identified three sex-specific genes: AKNA and *WHRN* were associated with total hip BMFF in males only, while *VMP1* was associated with spine BMFF in females only.

Furthermore, we tested more-stringent thresholds for functional annotation of the Meta-GWAS by sex groups. SNPs presented in Supplementary Data 48, 49 were based on a genome-wide significant *P*-value < $5 \times 10^{-8}$, with LD thresholds of $r^2 < 0.3$ for independent significant SNPs and $r^2 < 0.1$ for lead SNPs. The same LD $r^2$ thresholds were used in Supplementary Data 50, 51, but with a

genome-wide significant *P*-value < $1 \times 10^{-8}$. As for our meta-GWAS across both sexes, the use of more-stringent LD and P-value thresholds did not substantially alter the main findings (Supplementary Data 52), reinforcing the reliability of our results obtained using LD $r^2 < 0.6$ and *P*-value < $5 \times 10^{-8}$.

We also performed Sex x Genotype interaction analysis, which did not find P-sex genome-wide significant for any of the lead SNPs in four bone regions (Supplementary Data 53).

## Gene to function for sex-stratified meta-GWAS in the white population

MAGMA gene-set analysis results are displayed in Supplementary Figs. 22, 23. Most gene sets corresponded with those discovered in the BMI-adjusted, sex-combined meta-GWAS (Fig. 6), though there were some differences. Specifically, in females, total hip BMFF-associated genes were linked to nuclear receptors, laminopathies, and thyroid-related pathways, while no pathways were identified for genes associated with total hip BMFF in males. In each sex, spine BMFF-associated genes were linked to astrocytoma, kit pathway and SHP2 pathway. In females, spine BMFF genes were also linked to FLT3 signaling, STAT5 activation and MTOR signaling, while in males the spine BMFF genes were linked to T-cell receptor signaling (Supplementary Figs. 22, 23). MAGMA tissue expression analysis and cell-type-specific gene expression analysis are summarized in Supplementary Data 54, 55. In males or females, expression of genes associated with BMFF at each site was not significantly enriched among any specific tissue type. In BM single-cell expression analyses, we found that, in males, expression of genes associated with diaphyseal BMFF was significantly enriched in BM c-kit macrophages (C1qc high; *P* = $3.33 \times 10^{-4}$). We did not find any significant association that passed multiple testing correction for females. Using FUMA, we identified SNPs and mapped genes with

previously reported phenotypic associations in GWAS catalog. These showed overlap with the genomic risk loci we discovered in our meta-GWAS (Supplementary Data 56–59).

## Multi-ancestry meta-GWAS

We carried out the multi-ancestry meta-analysis using GWAS association summary statistics, yielding a total sample size of 44,514 in the femoral head, 44,441 in the total hip, 43,357 in the femoral diaphysis and 47,571 in the spine (Supplementary Data 3, 4, 6). GWAS results for the unrelated white population are summarized above. For GWAS in the non-white participants, sample quality control and the basic characteristics are summarized in Supplementary Data 5, 6, with further results reported in Supplementary Figs. 24–27. We observed minor genomic deflation in the non-white GWAS ($\lambda_{GC\_head} = 0.964$; $\lambda_{GC\_totalhip} = 0.967$; $\lambda_{GC\_diaphysis} = 0.967$; $\lambda_{GC\_spine} = 0.971$). Non-white GWAS identified three independent genome-wide significant signals for femoral head BMFF (rs114374538, rs13260214, rs13264172) across two loci mapped to *TNFRSF11B* and *COLEC10*, and three independent significant SNPs for diaphysis BMFF (rs9525641, rs922996, rs9533167) on chromosome 13 that mapped to *TNFSF11* (Supplementary Data 60). Two of the latter (rs9525641 and rs9533167) were also significant in the white population, whereas rs114374538, rs13260214, rs13264172, and rs922996 were significant only in the non-white population. No variants were associated at genome-wide significance in non-white GWAS for total hip or spine BMFF.

To further investigate genetic variants associated with different ethnic groups within the non-white population, we conducted GWAS analyses for Asian, Black, and mixed non-white ethnic groups (Supplementary Data 61–63). We found three, two, and ten independent significant SNPs for 'Asian', 'Black' and 'non-white mixed ethnic group', respectively (Supplementary Data 62). Their beta estimate directions were consistent as in the white population. However, these non-white ethnic subgroup GWAS analyses were found to be underpowered (Fig. 1C; Supplementary Data 64).

We then performed multi-ancestry meta-analyses of white and non-white populations for each bone region using MR-MEGA, which was designed for random effects models to penalize heterogeneity in different ancestries when combining effect estimates, allowing these estimates to be more generalizable across different ancestries[22]. In the multi-ancestry meta-analysis, the results for independent SNPs ($r^2 < 0.6$), lead SNPs ($r^2 < 0.1$), and mapped genes for each bone region are described in Supplementary Data 65–67. For femoral head BMFF, we identified 121 independent SNPs, 38 lead SNPs, and 65 mapped genes; total hip BMFF was associated with 314 independent SNPs, 86 lead SNPs, and 98 mapped genes; diaphysis BMFF was associated with 234 independent SNPs, 72 lead SNP, and 63 mapped genes; and for spine BMFF, we identified 310 independent SNPs, 80 lead SNPs, and 121 mapped genes. In addition, 46 (38.02%), 108 (34.39%), 65 (27.78%) and 94 (30.32%) significant SNPs in multi-ancestry meta-GWAS were found independent from those identified in the white population, for femoral head, total hip, diaphysis, and spine respectively (Supplementary Data 66). The direction of effect size for the significant associations for white and non-white populations remained consistent in each bone region (Supplementary Data 65–67).

## Gene to function for multi-ancestry meta-GWAS

We found 17 genes associated with BMFF in three bone regions (*LEPR, DNAJC13, NPHP3, ACAD11, NTSDC2, PBRM1, SM1M4, STAB1, TIMP4, PPARG, TERT, CCDC170, ESR1, COLEC10, TNFRSF11B, AKAP11, CYP19A1*), and 46 genes for BMFF in two bone regions (Supplementary Data 68). To identify the functional roles of BMFF-associated variants and which tissues and cells mediated the genetic effects, we performed MAGMA gene-set analysis, tissue expression analysis and cell-type-specific gene expression analysis (Supplementary Data 69, 70). The MAGMA gene-set analysis result was in line with that of meta-GWAS in the white

population. In MAGMA tissue expression analysis, no significant tissue associations were identified that surpassed the Bonferroni correction threshold for multiple testing for BMFF in the four bone regions (Supplementary Data 69). Cell-type-specific gene expression analysis showed the strongest association with genes expressed in BM mesenchyme fibroblasts ($P = 0.008$) for the femoral head BMFF, BM c-kit eosinophil progenitor cells ($P = 0.002$) for the femoral diaphysis BMFF, and BM mesenchyme endothelial cells (Ly6c1 high; $P = 0.029$) for the spine BMFF (Supplementary Data 70); however, these were not significant after Bonferroni correction. SNPs and mapped genes with previously reported phenotypic associations from published GWAS listed in the NHGRI-EBI catalog, which overlapped with the genomic risk loci identified in our meta-analyses, are summarized in Supplementary Data 71–74.

## Discussion

We have used our previously validated deep learning models to systematically quantify BM adiposity from MRI scans of over 45,000 participants in the UKBB imaging study. This is by far the largest BMFF analysis to date, being almost 100 times larger than the previous largest human studies[14,23]. The large scale of our analyses establishes reference ranges for BMFF at each site and extends understanding of the associations between BMFF and physiological parameters, including age, sex, BMD, BMI, and peripheral adiposity. The findings of lower BMFF in the spine than in the femoral sites, as well as the positive associations between BMFF at each site, are consistent with previous reports from us and others[12,14,16]. The age-dependent sex differences in spine BMFF also echo previous findings[17] and likely result from increased BMFF during menopause[2]. Importantly, we show that these age-dependent sex differences do not occur for femoral BMFF, which is higher in males than females regardless of age. Moreover, our study comprehensively identifies ethnicity-related differences in BMFF, which also vary according to sex and skeletal site.

In addition to these physiological insights, our findings have clinical implications. In particular, we definitively establish the robustness of the inverse association between BMFF and BMD at each site, highlighting the potential of BMFF as a biomarker for osteoporosis and fracture risk. We also show that spine BMFF is positively associated with BMI and peripheral adiposity, echoing previous reports of increased BMAT in obesity[2,19]. However, BMFF at each femoral site is inversely associated with BMI and shows more-complex relationships with other measures of peripheral fat mass that often differ between the sexes. This highlights complexities in the relationship between BMAT and peripheral adiposity, suggesting that the impact of BMAT on metabolic health may depend on the site of BMAT accumulation. BMFF's associations with BMD and peripheral adiposity may also explain our findings, including from cross-trait LDSC, which show that many BMFF-related genes are enriched among GWAS loci for traits such as BMD, osteoporosis, WHR, body composition, breast cancer, and cardio-metabolic diseases.

Our large-scale meta-GWAS identifies the overall as well as ancestry- and sex-specific genetic architecture for BMFF in the femoral head, total hip, femoral diaphysis, and spine. In meta-GWAS for the white population, we identified 67, 147, 134, and 174 independent association signals that mapped to 54, 90, 43, and 100 genes for the femoral head, total hip, femoral diaphysis, and spine respectively. Only one gene, *TIMP4*, was associated with all four bone regions, and ten genes were common to three bone regions (*LEPR, LEPROT, PPARG, TERT, CCDC170, ESR1, COLEC10, TNFRSF11B, TNFSF11, AKAP11*). We further found sex-specific BMFF-associated genetic variants, including two genes (*AKNA* and *WHRN*) associated with total hip BMFF only in males, and one gene (*VMP1*) associated with spine BMFF only in females. In GWAS for the non-white population, two genes (*TNFRSF11B and COLEC10*) were associated with femoral head BMFF, while one gene (*TNFSF11*) was associated with diaphysis BMFF.

The inflation in our meta-GWAS is consistent with polygenicity and batches one and two were found to share common genetic determinants. Our meta-GWAS also reveal a high degree of heritability for BMFF at each site, ranging from 19.99% for the femoral head, to 27.52% for the femoral diaphysis. This is similar to heritability for BMD (27.88%) and BMI (20.52%), and greater than heritability for WHR (13.85%) (Supplementary Data 36) or % body fat (17%)[24]. Thus, heritability of BMFF is similar to or greater than that for other adiposity- or bone-related traits.

Our findings identify a diverse range of genes associated with BMFF, many of which are implicated in biological phenomena that are of clear relevance to BM adiposity. *TIMP4*, the only gene associated with BMFF at all four sites, encodes the fourth member of the tissue inhibitors of metalloproteases (TIMPs) family and is particularly implicated in adipose and skeletal biology. TIMP4 is predominantly expressed in adipose tissue[25] and is linked to adipocyte development[26], lipid metabolism[27], and cartilage and bone remodeling and repair[28,29]. For example, mice lacking TIMP4 resist diet-induced obesity as a result of impaired lipid absorption[27]. These observations highlight several potential mechanisms through which *TIMP4* genetic variants might impact BMFF in humans. *TIMP4* genetic variants are also associated with body composition[30] and altered hematological traits[31]. The latter suggests that TIMP4 may influence hematopoiesis through its effects on BMAT. Given that *TIMP4* is the only gene mapping to all four BMFF sites, elucidating the underlying mechanisms through which *TIMP4* impacts BM adiposity should be a priority for future research.

The second and third common loci found for femoral head, total hip, and spine BMFF mapped to the leptin receptor (*LEPR*) and leptin receptor overlapping transcript (*LEPROT*). LEPR is a marker of most bone marrow stromal cells and osteogenic lineage cells in adult long bones[32,33] and is particularly highly expressed in skeletal stem cells that are primed toward adipogenesis, at least in mice[34,35]. The hormone leptin is well known for its key roles in regulating body weight and energy balance, but it also impacts bone biology through direct and indirect mechanisms[36]. Therefore, *LEPR* and *LEPROT* genetic variants may influence BMFF by modulating the fate of BM progenitors, impacting energy balance, and/or by altering leptin's direct and indirect skeletal effects.

The fourth notable association, common to total hip, diaphysis, and spine BMFF, is for *PPARG*. This gene encodes peroxisome proliferator-activated receptor gamma (PPARγ), a ligand-activated nuclear receptor that acts as the master regulator of adipocyte differentiation[37]. This includes BMAT formation, which is suppressed by genetic or pharmacological inhibition of PPARγ[38] but stimulated by thiazolidinediones, synthetic PPARγ agonists[39,40]. *PPARG* genetic variants are also associated with WHR, type 2 diabetes, and some hematological traits[30,31,41]. Our study extends these previous observations by showing that variation in the *PPARG* gene is also associated with altered BM adiposity.

The associations with *TERT*, observed for femoral head, total hip, and spine BMFF, are also intriguing in light of previous studies. *TERT* encodes telomerase reverse transcriptase, a component of telomerase, which influences cell division and senescence. Previous GWASes identify associations between *TERT* and various cancers[42], but the strongest associations are for hematological traits[30,31]. *TERT* mutations have also been implicated in BM failure[43,44]. Therefore, an intriguing possibility is that *TERT* genetic variants influence BM function and hematopoiesis, in part, by altering BM adiposity. Senescence within adipose tissues also impacts adipose formation and function[45], suggesting that *TERT* genetic variants may modulate BM adiposity through effects on BM senescence.

The sixth significantly associated locus, common to femoral head, total hip, and diaphysis BMFF, mapped to the *ESR1* gene, encoding estrogen receptor-α. This receptor is expressed in breast cancer tissues and associated with breast cancer risk[46], consistent with our MAGMA results showing that breast-cancer-related pathways are associated with BMFF at these sites. More pertinently, numerous studies have established that estrogen suppresses BM adiposity in animals and humans[2]. Indeed, BM adiposity increases in menopause or ovariectomy and this is suppressed by exogenous estrogen treatment[2]. Variation in estrogen exposure also contributes to natural fluctuations in BMFF during the human menstrual cycle[47]. Moreover, estrogen acts via ERα to inhibit bone resorption and stimulate bone formation[48], and *ESR1* genetic variants are associated with BMD and body fat distribution in humans[49,50]. Therefore, the association between *ESR1* and femoral BMFF may relate to estrogen's actions on skeletal remodeling and/or fat partitioning.

Two other loci common to femoral head, total hip, and diaphysis BMFF mapped to *TNFSF11*, encoding the receptor activator of nuclear factor kappa B ligand (RANKL), and *TNFRSF11B*, which encodes osteoprotegerin (OPG). Related to these genes, loci common to total hip and diaphysis BMFF mapped to *TNFRSF11A*, which encodes RANK, the RANKL receptor. *TNFRSF11A* is notable because it was also identified through our TWAS and colocalization analyses for diaphysis BMFF and visceral adipose eQTLs. Through RANK, RANKL activates osteoclasts to stimulate bone resorption, while OPG is a secreted decoy receptor that binds to RANKL and thereby inhibits its pro-resorptive effects[51]. These three genes were among the first GWAS hits for BMD and osteoporosis[49], suggesting that they influence BMFF via their effects on skeletal remodeling. Indeed, osteoporosis therapies that inhibit bone resorption, such as bisphosphonates, decrease BMAT[52]; whether this occurs for denosumab, a clinical RANKL inhibitor, remains to be determined[53]. BM adipocytes and their progenitors also express RANKL[2,54], suggesting that genetic variation in *TNFSF11*, *TNFRSF11B* and *TNFRSF11A* might alter BMFF by directly affecting BM adipocyte formation and function.

The ninth common locus found for femoral head, total hip, and diaphysis BMFF, mapped to *AKAP11* (A-Kinase Anchoring Protein 11), which is adjacent to the *TNFSF11* gene. AKAP11 belongs to a group of scaffolding proteins that attach to the regulatory subunit of protein kinase A[55]. Previous GWASes show that *AKAP11* genetic variants are associated with BMD, osteoporosis, and arthritis[49], as well as blood cell counts[30,31,41]. Therefore, one possibility is that AKAP11 influences skeletal remodeling and hematopoiesis, in part, by modulating BM adiposity.

The above genes each encode proteins with well-established biological functions. In contrast, *COLEC10* and *CCDC170*, associated with femoral head, total hip, and diaphysis BMFF, are less well studied. *COLEC10*, encoding collectin subfamily member 10, is involved in the lectin complement pathway, which may coordinate cell migration and organ formation during embryogenesis. COLEC10 mutations are associated with disorders of craniofacial development[56]. Previous studies found that variants mapping to this gene are associated with BMD, blood pressure, and hematological traits[30,31,49].

*CCDC170* is especially intriguing because it was also identified in our TWAS analyses, suggesting direct connections between BMFF-associated genetic variants and altered *CCDC170* expression. The CCDC170 protein might be linked to the Golgi apparatus and protein glycosylation[46], and CCDC170 is co-expressed with ESR1 in breast cancer tissues[57]. Previous GWASes identified associations between the *CCDC170-ESR1* locus and breast cancer risk[58]. Similarly, *CCDC170* is associated with BMD[49] and body fat distribution[50]; however, the function of CCDC170 in bone or adipose biology is still unclear. One study found that knockdown of CCDC170 in mice suppresses bone formation, but the effects on BMAT were not assessed[59]. Our meta-GWAS and TWAS results are consistent with these studies and highlight the need for future research to elucidate the interplay between CCDC170, BM adiposity, adipose distribution, and skeletal health.

The identification of three sex-specific BMFF-associated genes, *AKNA, WHRN,* and *VMP1*, is reminiscent of the sexually dimorphic genetic associations identified for body composition and other related traits[50]. *AKNA* encodes the AT-hook transcription factor, which is most highly expressed in BM and lymphoid tissues[25] and regulates B-lymphocyte development[60]. Intriguingly, AKNA is also a centrosomal protein that regulates microtubule organization. Cytoskeletal dynamics directly influence adipogenesis and osteogenesis[61], suggesting that *AKNA* genetic variants may directly impact BM adipogenesis. However, the mechanisms allowing AKNA's dual roles in transcription and cytoskeletal function remain unclear, and any direct impact on skeletal stem cell fate remains to be investigated.

*WHRN* encodes whirlin, a protein implicated in sterocilia elongation and actin cystoskeletal assembly. *WHRN* mutations occur in Usher syndrome, a neurosensory disorder affecting hearing and vision[62]. Although primary cilia are involved in adipogenesis and white adipose tissue expansion[63], stereocilia have not yet been linked to these phenomena. Instead, WHRN may impact BM adiposity through tissue-extrinsic mechanisms. Indeed, WHRN expression is greatest in the adrenal glands[25,64], which is notable because glucocorticoids increase BM adiposity[2] and exert sexually dimorphic effects[65]. Therefore, one possibility is that WHRN influences BMFF in males, but not females, by modulating glucocorticoid exposure.

Finally, *VMP1* encodes vacuole membrane protein 1, a phospholipid scramblase involved in lipid homeostasis, membrane dynamics, and autophagy[66]. Autophagy directly influences white adipose tissue development and function[67], and VMP1 has relatively high expression within the BM[25]. Thus, we speculate that VMP1 may directly impact BM adiposity by modulating autophagy and lipid metabolism within the BM.

Notably, none of these three genes is differentially expressed between males and females, at least among tissues profiled in GTEx[68]; it remains unknown if their expression in bone or BM, which are not included in GTEx, differs between the sexes. These genes' sexually dimorphic effects on BMFF may also result from interactions with sex steroids and/or other physiological phenomena that differ between males and females. Considering the growing interest in sex differences, these remain important questions for future research.

The genes discussed above are those associated with BMFF at three or more skeletal sites; another 217 genes were associated with only one or two sites. Although these are too numerous to discuss herein, this site-specific nature of the genetic associations further supports the concept that BMAT formation and function differs according to skeletal site[2,7]. This is true not only for spine vs femoral BMAT, but also for BMAT at the different femoral sites, which each show distinct genetic associations (Fig. 5A). Together with our findings from cross-trait LDSC, the number and diversity of these BMFF-associated genes should allow construction of polygenic risk scores for BMFF. One promising possibility will be to leverage these as genetic biomarkers for precision medicine and other translational applications.

Our MAGMA gene ontology, TWAS, and co-localization analyses also provide further insights from across all BMFF-associated loci. Unexpectedly, for each site, MAGMA revealed that genes associated with BMFF were enriched in breast-cancer-related pathways. This likely reflects the common association of these sites with genes implicated in this disease, including those discussed above (*TIMP4, PPARG, ESR1, TERT, CCDC170*) and those associated with only one or two sites (e.g., *GDF5* for femoral head and total hip BMFF; *FAM189B, SCAMP3* and *CLK2* for spine BMFF). However, it might also indicate direct connections between BMFF and breast cancer pathogenesis. Indeed, one small study identified BMFF as a predictor of breast cancer risk[69], and breast cancer cells preferentially metastasize to BMAT-rich skeletal niches[70]. The unexpected link between diaphysis BMFF genes and mammary gland development, which involves common associations with *ESR1, TNFSF11,* and *TNFSFR11A*, may also relate to such putative BMAT-breast cancer relationships. The other MAGMA-identified pathways relate to skeletal biology or disease, stem cell function, or hematology, phenomena that each have more-obvious relevance to BM adiposity.

Our TWAS and colocalization analyses help to refine the GWAS results by identifying connections between BMFF and altered gene expression. Among the TWAS results, *UQCC1* and *NT5DC2* are notable because they were the only genes whose mesodermal expression was associated with genetic variants for BMFF at three sites: the femoral head, total hip, and diaphysis. The other TWAS hits were linked to BMFF at only one or two sites, but several of these were also identified through colocalization analysis; these include *TNFRSF11A, CCND2, CYP19A1, EEFSEC,* and *STXBP6*. It will be important to determine how these genes might impact BM adiposity, including whether this occurs through BM-intrinsic or –extrinsic mechanisms.

A limitation to our TWAS and colocalization studies is that they rely on gene expression data from GTEx, which lacks bone or BM[64]. Although BMAT shares some transcriptional characteristics with the mesodermal and lymphoid tissues assessed in our TWAS and colocalization analyses, it is also clear that BM and BMAT have a transcriptomic profile that is distinct from these other tissues[71–73]. This reliance on GTEx data is also a limitation of our MAGMA tissue expression analysis, which found that, for each site, the BMFF-associated genes are not enriched in any specific tissues. One interpretation of this is that the BMFF-associated genes influence BM adiposity via tissue-extrinsic mechanisms, such as the systemic endocrine and metabolic changes that can influence BMAT formation[2]. However, our cell expression analyses verified that the BMFF-associated genes are enriched in specific BM cell types, including mesenchymal fibroblasts, BM c-kit macrophages (C1qc-high), BM c-kit eosinophil progenitor cells, and BM mesenchymal endothelial cells (Ly6c1high) for the femoral head, total hip, diaphysis, and spine respectively. We demonstrate heritability enrichment in relevant gene-sets and cell types, suggesting that BM-intrinsic mechanisms also contribute to BMAT formation and function. Ultimately, it is likely that the effects on BMFF include both BM-intrinsic and -extrinsic mechanisms, as well as complex interactions between these. Our identification of BMFF-associated genes, including those linked to altered expression in mesodermal and/or lymphoid tissues, provides a robust foundation for future studies to comprehensively investigate these mechanisms.

There are several limitations to our study. First, the UKBB abdominal MRI protocol uses dual-echo sequences, which do not allow precise T2* correction to be applied to fat fraction measurements. The presence of trabecular bone might bias BMFF measurements by causing T2* decay effects, which can vary in the water and fat components. Thus, these dual-echo data cannot be used to determine the adjusted proton density fat fraction[10]. However, as highlighted previously[14], the substantial size of the UKBB cohort, as well as the use of standardized and optimized MRI protocols across all imaging centers involved, are likely to limit biases relating to such T2* effects, yielding BMFF values comparable to the proton density fat fraction. Another limitation of the UKBB MRI data is that deviations in MRI acquisition or data mislabeling can impair segmentation of target BM sites, especially the femoral head and total hip. However, we show that such faulty segmentations are robustly identified and excluded during our error-checking process and account for <0.15% of deep learning outputs[14] (Supplementary Fig. 28). Therefore, our deep learning models produce robust BMFF measurements. A third limitation relates to the nature of the UKBB cohort, which consists primarily of older white participants. Thus, our findings in populations of European ancestry might not generalize to other ethnic groups. Although we conducted GWAS in non-white UKBB participants, this non-white GWAS was limited in scale and underpowered compared to our white population meta-GWAS. This is reflected in the low SNP heritability and small number of genome-wide significant findings in the non-white GWAS. Stratification of the non-white participants into Asian, Black,

and mixed backgrounds showed that GWASes for these ethnicities were also underpowered (Fig. 1C; Supplementary Data 64). Furthermore, these results are prone to bias as a self-reported ethnicity parameter was used. The lack of younger UKBB participants (<50) years old) also impairs the identification of age-related increases in BM adiposity and the discovery of genes that may influence BMFF at younger ages. We suggest future GWASes increase the diversity of ancestries and ages, which could advance our understanding of the genetic susceptibility to BMFF for all populations. Fourth, our meta-GWASes focused on investigating common genetic effects. We recognize that variants with MAF < 1% in one or more populations, as well as indels and structural variants, may contribute to the observed associations. Nevertheless, we adhered to the established GWAS protocols using variants that passed stringent quality control. Future work could consider including imputation using diverse reference panels from long-read sequence data to improve genomic coverage, which would help to identify more and lower-frequency variants, providing insights into structural variation that may be population-specific. Lastly, we were not able to validate our meta-GWAS results in a genetically similar cohort. However, we ran the GWAS in two batches of BMFF measurements and reported the meta-GWAS results, where we found the direction of effect sizes to remain consistent across the two batches. The UKBB currently is the only large-scale study providing MRI data for the bones and joints, which is required for BMFF analysis. We seek to validate our findings in the remaining ~50,000 participants of the UKBB imaging study once these data are available; however, the provision of similar MRI data from other large-scale genetic studies should be prioritized to further advance understanding of the genetics and pathophysiology of BM adiposity.

In conclusion, our deep learning models have allowed the largest BMFF analysis to date. Our large-scale meta-GWAS identifies genetic variants that influence BM adiposity at multiple skeletal sites, and we establish the site-specific relationships between BMFF and other body composition traits relevant to skeletal and cardiometabolic health. These findings shed light on potential molecular mechanisms that influence BMAT formation and function and open new avenues for future studies, including Mendelian Randomization and translational application of genetic BMFF biomarkers, to comprehensively establish how BMAT impacts human health and disease.

## Methods

### Ethics
The research reported herein was done in compliance with all ethical requirements. Data for the present study were obtained under an approved UKBB project application (ID 48697), which provided specific approval to measure BMFF from the UKBB MRI data. UKBB has ethics approval from the National Health Service North-West Center Research Ethics Committee (Ref: 11/NW/0382). All UKBB participants provided their consent to take part in the UKBB study, The UKBB is conducted in accordance with criteria set by the Declaration of Helsinki, with participants providing informed written consent to take part and to be followed up through national record linkage.

### Study population
UKBB is a large population-based prospective cohort study of 502,352 participants aged 40–60 years recruited between 2006 and 2010[74]. In this study, we used participant data from the ongoing UKBB multi-modal imaging study, initiated in 2014, which comprises brain, cardiac, and abdominal MRI, carotid ultrasound scan, and whole-body dual-energy X-ray absorptiometry[15].

### Deep learning for BM segmentation and BMFF measurement from Dixon MRI data
We applied our previously developed and validated deep learning models[14] to the UKBB abdominal MRI data to measure BMFF of the

femoral head, total hip, femoral diaphysis and spine. Briefly, we developed and validated a light-weight attention-based U-Net model for simultaneous detection and segmentation of tiny structures in large 3D MRI imaging data, to generate volumes of interest (VOIs) corresponding to BM regions. We identified and excluded deep learning segmentation outliers, including those with abnormally small segmentation volumes. Detailed sample quality control is presented in Supplementary Data 1 and Supplementary Data 2. The segmented VOIs were then applied to the fat fraction (FF) maps. Further details of the deep learning model and application are in our previously published paper[14]. The deep-learning BMFF measurements were conducted in two batches based on the availability of MRI data released by UKBB.

### Principal component analysis and MRI image error checking to identify technical outliers from deep learning segmentation
During our development and validation of these deep learning models we found that, in rare cases, segmentation is compromised because the target regions of interest (ROIs) do not fall within the expected imaging volumes[14], yielding incorrect BMFF values. To identify such outliers in the present study, principal component analysis (PCA) was carried out on normalized BMFF measures at each skeletal site in a total of 46,717 participants. The outcome of the PCA was visualized using Score plots, which revealed two distinct clusters of participants: a major cluster comprising most participants, and a minor cluster in the bottom right of the plot, comprising <100 participants (Supplementary Figs. 28A–D). To determine how these clusters relate to BMFF, we shaded the plots according to the BMFF% of each skeletal site. This revealed that the bottom-right minor cluster consisted of participants with abnormally low femoral head and/or total hip BMFF of below 60% (Supplementary Figs. 28B, C), whereas BMFF values for the spine or diaphysis were similar to those for the major cluster (Supplementary Fig. 28A, 2D). Further inspection of this minor cluster confirmed that a total of 89 participants presented with BMFF below 60% at these sites: 71 participants had femoral head and total hip BMFF below 60% and 18 participants had BMFF below 60% for either the femoral head or total hip. The abnormality of these very low BMFF values was further apparent in histograms of BMFF distribution at each site (Supplementary Fig. 28E, F), in which there was a small, atypical peak of values < 60%.

We then investigated if these very low BMFF values result from technical issues or if they represent genuine biological outliers. To do so, we manually inspected the MRI Dixon images of these 89 participants to determine if the low femoral head and/or total hip BMFF values are a result of technical issues or if they represent individuals with biologically low BMFF at these sites. A typical MRI volume in which these two ROIs are completely contained is shown in Supplementary Fig. 28G. In contrast, most of the very low BMFF values for these two regions were a result of the MRI volumes containing incomplete ROIs (Supplemental Fig. 28H). In these cases, femoral head and total hip ROIs were either split across two MRI volumes (or the remaining part of the ROI in one image folder was not found in any other folder), or the UKBB source data were mislabelled so that our deep learning models were presented with water fraction images instead of the required fat fraction images. Such cases, which were classified and excluded as technical outliers, accounted for ~93% of participants with BMFF < 60%, and comprised most of the distinct minor cluster seen in Supplementary Fig. 28A–D. However, several participants had femoral head and/or total hip BMFF < 60% despite having completely captured ROIs at these sites (Supplementary Fig. 28I): the ROIs were similar to those from typical participants (Supplementary Fig. 28G) but had much weaker fat signal. These participants were considered to have biologically low BMFF at these sites, rather than being technical outliers, and therefore were included in the final dataset used for GWAS analysis. The Score plot from the PCA performed on this final dataset (excluding the technical outliers,

$n = 46,633$) is shown in Supplementary Fig. 28J: the minor cluster is no longer present, and instead those participants with biologically low femoral head and/or total hip BMFF are scattered among the periphery of the single cluster.

## BMFF descriptive and association analysis

BMFF % are presented as violin plots (median: dashed horizontal line; quartiles: solid horizontal lines) for each bone region. We compared our BMFF measurements with the UKBB-provided MRI-based measurements of visceral adipose tissue volume (VAT) and abdominal subcutaneous adipose tissue volume (ASAT), which were divided by height$^2$ (m$^2$) for each participant to generate the VAT index (VATi) and ASAT index (ASATi) (Supplementary Data 9). We also compared our BMFF measurements with the UKBB-provided DXA-based measurements of total and regional fat % (Supplementary Data 9). Statistical significance for BMFF between age and sex groups was assessed using multivariate ANOVA tests, with Tukey's multiple comparisons test to compare white and non-white participants for each site. Multivariable linear regression models were fitted to test the association between rank-transformed (normalized) BMFF and BMD, adjusting for age, sex, and BMI at each site (Fig. 3)[75]. Statistical analyses were performed using R (version 4.1.2) and Prism (v10.1.1, GraphPad, USA).

## Genome-wide association analyses

We used the UK Biobank imputed genotypes version 3[76]. Genotyping, quality control and genotype imputation were conducted by UK Biobank, Wellcome Trust Center for Human Genetics (WTCHG), University of Oxford, UK[76]. To summarize, the initial 50,000 participants were genotyped by the Affymetrix UK BiLEVE Axiom array and the subsequent 450,000 participants were genotyped by the Affymetrix UK Biobank Axiom array. Genotype imputation was performed using the Haplotype Reference Consortium (HRC) panel, the UK10K panel, and the 1000 Genome Phase 3 panel, as the reference panel. Heterozygosity outliers, missing rate outliers, sex mismatches and individuals with sex chromosome aneuploidy were excluded after quality control[76]. We retained only high-quality SNPs with missingness <0.05, Hardy-Weinberg equilibrium (HWE) test $P$-value > $10^{-12}$, non-multiallelic, imputation quality (INFO) > 0.4, and minor allele frequency (MAF) > 0.005. Family relatedness was controlled for using kinship coefficients derived by UK Biobank, which identify related individuals up to a 3$^{rd}$ degree. One individual from each cluster of related individuals was retained based on data availability. We sub-grouped participants based on their ancestry as 'white' and 'Non-white' using UKBB data-field 22006 ("Genetic ethnic grouping"). 'Non-white' was further categorized into 'Asian', 'Black' and 'non-white mixed ethnic group' by data-field 21000 ("Ethnic background", as self-reported by participants).

GWAS analyses to investigate associations between imputed genotypes and BMFF were adjusted for age at imaging visit, sex, BMI at imaging visit, genotyping batch, and population structure of the first 40 principal components (PCs 1-40). These analyses were done by creating rank-transformed BMFF residuals and then regressing them against HRC-imputed genotype dosages using RegScan v0.5[77]. Sensitivity analysis was performed without BMI adjustment, to investigate variant-specific pleiotropic associations regarding the impact of BMI adjustment on genetic effect sizes. We conducted ancestry- and sex-specific GWAS analyses for BMFF of the first and the second batch for the four bone regions including femoral head, total hip, femoral diaphysis, and spine respectively. We performed GWAS power calculation using GCTA package (https://yanglab.westlake.edu.cn/software/gcta/#GREMLpowercalculator). Genome-wide significance was defined as $P$-value < $5 \times 10^{-8}$. GWAS summary statistics were visualized using circular Manhattan and QQ plots[78]. The λ estimate was calculated to evaluate genomic inflation[79].

## GWAS meta-analysis of BMFF

We conducted meta-analyses of GWASes in the first and second BMFF batches for white population under IVW fixed effects models for the four bone regions respectively, using META v1.7, to improve the power to detect associations and to cross-compare or replicate associations across the two batches of BMFF measurements[80]. The I$^2$ statistics were calculated to quantify the degree of heterogeneity in allelic effects between the two GWASes at each variant[81]. Meta-analyses results were further filtered to exclude any variants with I$^2$ > 65%[79]. To investigate sex-specific genetic associations, we further performed sensitivity meta-analyses of white population for the two batches in male and female groups. We also analyzed the Sex x Genotype interaction, using the following formula, developed by Winkler et al.[82], to calculate t-sex and P-sex:

$$t_{sex} = \frac{b_M - b_F}{\sqrt{SE_M^2 + SE_F^2 - 2r_{sex} \times SE_M \times SE_F}}$$

In addition, we conducted sensitivity analysis without BMI adjustment for meta-GWAS in the white population, both for the sexes combined and when stratified by sex.

We performed multi-ancestry meta-analyses of white and non-white populations for each bone region using the Meta-Regression of Multi-Ethnic Genetic Association (MR-MEGA v0.20)[83]. MR-MEGA performs multi-ancestry meta-regression to model allelic effects, derived from mean pairwise allele frequency differences[22]. To quantify heterogeneity in allelic effects between populations, variants with residual heterogeneity P value at nominal significance ($P < 0.05$) were excluded[83].

## Genomic risk loci and functional annotation

We used FUMA v1.5.2 (http://fuma.ctglab.nl) for functional annotation of the Meta-GWAS-white and MR-MEGA summary statistics. We used 1000 Genomes phase 3 (European population for Meta-GWAS-white; all populations for MR-MEGA) as the linkage disequilibrium (LD) reference population. FUMA identified genome-wide significant SNPs ($P$-value < $5 \times 10^{-8}$) in low LD with each other (r$^2$ < 0.6) as independent significant SNPs and further subdivided these into independent lead SNPs if they are in approximate linkage disequilibrium with each other (r$^2$ < 0.1). Genomic loci were then defined using the LD blocks of independent significant SNPs, where an LD block is the region containing all the SNPs in LD (r$^2$ ≥ 0.6) with the independent significant SNP. In cases where the LD blocks of multiple independent significant SNPs are in close physical proximity to each other (within 250-kb), the LD blocks were merged into a single genomic locus[84]. In addition, we tested more-stringent thresholds for functional annotation of the Meta-GWAS-white and MR-MEGA summary statistics: these more-stringent thresholds include a $P$-value < $1 \times 10^{-8}$ to identify genome-wide significant SNPs[85], among which an LD threshold of r$^2$ < 0.3 was used to identify independent significant SNPs and r$^2$ < 0.1 to define lead SNPs.

## Gene mapping

The individual genomic risk loci were mapped to genes using the "SNP2GENE" function in FUMA[20]. Positional mapping was performed based on ANNOVAR annotations, applying a maximum distance of 10 kb between SNPs and genes. A Bonferroni-corrected significance threshold (adjusting for 19,175 protein-coding genes) of $P$-value < $2.608 \times 10^{-6}$ was set for the gene-based GWAS[86].

## MAGMA gene-set analysis and MAGMA tissue expression analysis

MAGMA gene-set analysis was performed where variants map to 15,496 gene sets and GO terms from the MSigDB v.7.0 database in

FUMA[21]. MAGMA tissue expression analysis was performed using GTEx v8's 54 tissue-type gene expression profiles in FUMA[87]. Gene set and tissue expression analyses were Bonferroni corrected[20,21].

## Cell-type-specific expression analysis

MAGMA gene-property analysis was performed to calculate associations between gene-wise $P$ values from the Meta-GWAS and cell-type-specific gene expression in FUMA. Bone marrow-cell-type expression data were drawn from single-cell RNA-seq (scRNA-seq) data from mouse bone marrow. For each gene, the value for each cell type was calculated by dividing the mean unique molecular identifier (UMI) counts for the given cell type by the summed mean UMI counts across all cell types[20].

## SNP heritability and genetic correlation

We used linkage disequilibrium score regression (LDSC: https://github.com/bulik/ldsc) to estimate genomic inflation and SNP-based heritability ($h^2_{SNP}$)[88]. Precomputed LD scores from the 1000 Genomes European reference population were used (https://data.broadinstitute.org/alkesgroup/LDSCORE/eur_w_ld_chr.tar.bz2). Genetic correlations ($r_g$) between the signal from GWASes in the first and second BMFF batches were also calculated using LDSC[88] using a previously described statistical framework[89]. Briefly, $r_g$ was calculated by normalizing the genetic covariance $\rho_g$ by the estimated SNP heritabilities for the two traits: $r_g = \frac{\rho_g}{h^2_{g1} x h^2_{g2}}$, where $h^2_{g1}$ and $h^2_{g2}$ are the SNP heritabilities for trait Y1 (BMFF_1st batch) and $Y_2$ (BMFF_2nd batch), respectively.

## Cross-trait LD score regression

We investigated genome-wide sharing of common variants between BMFF and BMD, fat-related traits (BMI, WHR), and disease traits (such as osteoarthritis, type 2 diabetes, coronary artery disease, stroke, hypertension, breast cancer), using cross-trait LDSC[79]. We checked GWAS catalog (https://www.ebi.ac.uk/gwas/) to identify the most recently published meta-GWAS with the largest sample size for each BMD, fat-related traits, and disease traits to compare with our meta-GWAS for BMFF in the white population. Cross-trait LDSC estimated the genetic correlation ($r_g$) between two traits. The slope (coefficient) represents genetic correlation ($r_g$). The estimated range of the LDSC $r_g$ is from −1 to 1, where −1 indicates an absolute negative genetic correlation and 1 indicates an absolute positive genetic correlation[90]. The statistical framework for cross-trait LDSC was as described previously[91]. In brief, cross-trait LD score regression was calculated based on the formula:

$$E[z_{1j}z_{2j}] = \frac{\sqrt{N_1 N_2}\rho_g}{M}\ell_j + \frac{\rho N_s}{\sqrt{N_1 N_2}}$$

where $z_{ij}$ denotes the z-score for study $i$ and SNP $j$, $N_i$ is the sample size for study $i$, $\rho_g$ is genetic covariance, $\ell_j$ is LD Score[92], $N_s$ is the number of individuals included in both studies, and $\rho$ is the phenotypic correlation among the $N_s$ overlapping samples.

## Transcriptome-wide association studies (TWAS)

We conducted TWAS analysis using FUSION, by integrating gene expression prediction models generated from subcutaneous adipose tissue, visceral-omentum adipose tissue and skeletal muscle tissue (GTEx v8) with meta-GWAS for BMFF in the white population to evaluate associations between genetically predicted gene expression and BMFF risk.

TWAS incorporated BMFF meta-GWAS summary statistics into cis-eQTL information representing the relationship between SNPs and gene expression in the specific tissues and accounted for LD to identify candidate genes associated with traits. Pre-computed gene expression weights from GTEx v8 for adipose and skeletal muscle tissues were

used as downloaded from the FUSION. TWAS was performed using a LD reference panel based on the 1000 Genomes Project's samples of European ancestry[93]. FUSION calculated GE using a linear mixed model such as LASSO, elastic net, and Bayesian sparse linear mixed model (BSLMM)[94]. FUSION prioritized the model with the highest 5-fold cross-validated performance for each gene, and the optimized results were displayed.

The transcriptome-wide significance threshold for the TWAS associations in this study was Bonferroni corrected. The TWAS $Z$-score plot was generated using a TWAS-plotter function (https://github.com/opain/TWAS-plotter).

## Colocalization

We performed colocalization to determine whether the same genetic variant was responsible for both an eQTL effect (change in gene expression) and a meta-GWAS signal (BMFF trait). We used the common variants in our meta-GWAS-white and eQTL summary statistics corresponding to the gene-expression references in mesodermal and lymphoid tissues (subcutaneous adipose tissue; visceral omentum adipose tissue; skeletal muscle tissue; spleen tissue; and Epstein Barr Virus (EBV)-transformed lymphocytes) from GTEx v8. Colocalization analysis estimates posterior probabilities for five hypotheses: H0: no association, H1: GWAS association only, H2: eQTL association only, H3: Both are associated but not colocalized, H4: Both are associated and colocalized. A posterior probability of hypothesis 4 (PPH4) measures the probability that a locus is colocalized as the result of a single causal variant, as opposed to two distinct causal variants (PPH3). Loci with PPH4 > 80% were considered as colocalized. To select genes for testing, we mapped independent significant SNPs using Variant Effect Predictor[95]. Colocalization was performed using 'coloc.abf' function from Coloc R package[96].

## Reporting summary

Further information on research design is available in the Nature Portfolio Reporting Summary linked to this article.

## Data availability

All data for BMFF and BM segmentation volumes have been uploaded to the UKBB (upload ID 5858), where they will be available to any individuals with an approved UKBB project. Researchers can apply for UKBB access via the UKBB Access Management System (https://ams.ukbiobank.ac.uk/ams/). Data used for LDSC GWAS were obtained from GWAS catalog. Pubmed IDs and URLs for the relevant studies from GWAS catalog are presented in Supplementary Data files 32-36, 56-59, and 71-74. For TWAS, pre-computed gene expression weights from GTEx v8 for adipose and skeletal muscle tissues were used as downloaded from the FUSION. The remaining data are reported in the Supplementary Data files.

## Code availability

Details of all code used for GWAS analyses, LDSC, TWAS, Colocalization, and FUMA, is described above and in the reporting summary; where relevant, details are also reported in the legends for the tables in the Supplementary Data files. Python code for deep learning segmentation of bone marrow volumes in the spine, femoral head, total hip, and femoral diaphysis is available on Zenodo under https://doi.org/10.5281/zenodo.13959673 and on Github at https://github.com/chengjiawang/OPTIMAT_NET/tree/iniRelease. Matlab code for sorting UK Biobank MRI data (prior to segmentation) and for fat fraction mapping is available at on Zenodo under https://doi.org/10.5281/zenodo.13961316 and on Github at https://github.com/WillCawthorn/OPTIMAT.

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

## Acknowledgements

This work was supported by grants from the Medical Research Council (MR/S010505/1 to W.P.C. and E.T.), the British Heart Foundation (RE/18/5/34216 for salary support to W.X.; RG/16/10/32375 to support C.W.; 4-year PhD studentship FS/4yPhD/F/22/34175 C for S.S.), Cancer Research UK (Career Development Fellowship C31250/A22804 to E.T.), the University of Edinburgh (Chancellor's Fellowship to W.P.C.), and the Edinburgh Clinical Research Facility and NHS Lothian R&D (funding to C.G. and T.M). We are grateful to Dominic Job (Edinburgh Imaging, University of Edinburgh) for support with IT infrastructure, including GPU servers. Rights Retention Statement: For the purpose of open access, the authors have applied a Creative Commons Attribution (CC-BY) license to any Author Accepted Manuscript version arising from this submission.

## Author contributions

Conceptualization, W.P.C.; Data curation, W.X., I.M.E., D.M.M., C.W., S.S., G.P. and W.P.C.; Formal Analysis, W.X., I.M.E., D.M.M., C.W., S.S., G.P., C.D.G. and W.P.C.; Funding Acquisition, S.M.F., M.G.D., S.I.S., T.M., E.T. and W.P.C.; Investigation, W.X., I.M.E., D.M.M., C.W., S.S., G.P., E.T. and W.P.C.; Methodology, W.X., I.M.E., D.M.M., C.W., S.S., G.P., C.D.G., S.B., J.P., X.L., P.R.H.J.T., M.T., S.I.S., T.M., E.T. and W.P.C.; Project administration, S.I.S., T.M., E.T. and W.P.C.; Resources, S.I.S., T.M., E.T. and W.P.C.; Software, D.M.M., C.W., G.P.; Supervision, S.I.S., T.M., E.T. and W.P.C.; Visualization, W.X., D.M.M., C.W., C.D.G., S.S. and W.P.C.; Writing – Original Draft, W.X., S.S., E.T. and W.P.C..; Writing – Review & Editing, W.X., I.M.E., G.P., J.P., M.T., E.T. and W.P.C.

## Competing interests

G.P. is currently an employee of Pfizer; however, Pfizer had no role in the design or interpretation of this research. All other authors declare no competing interests.

## Additional information

[1]Centre for Global Health and Molecular Epidemiology, Usher Institute, University of Edinburgh, Edinburgh, UK. [2]University/BHF Centre for Cardiovascular Science, University of Edinburgh, The Queen's Medical Research Institute, Edinburgh BioQuarter, 47 Little France Crescent, Edinburgh, UK. [3]Edinburgh Imaging, University of Edinburgh, The Queen's Medical Research Institute, Edinburgh BioQuarter, 47 Little France Crescent, Edinburgh, UK. [4]School of Mathematics and Computer Sciences, Heriot-Watt University, Edinburgh, UK. [5]Archimedes Unit, Athena Research Centre, Marousi, Greece. [6]Univ. Lille, CHU Lille, Marrow Adiposity and Bone Laboratory (MABlab) ULR 4490, Department of Rheumatology, Lille, France. [7]Department of Big Data in Health Science, School of Public Health and The Second Affiliated Hospital, Zhejiang University School of Medicine, Hangzhou, China. [8]Medical Research Council Human Genetics Unit, Medical Research Council Institute of Genetics & Molecular Medicine, University of Edinburgh, Edinburgh, UK. [9]Danish Institute for Advanced Study (DIAS), Epidemiology, Biostatistics and Biodemography, Department of Public Health, University of Southern Denmark, Odense, Denmark. [10]Cancer Research UK Edinburgh Centre, Medical Research Council Institute of Genetics and Cancer, University of Edinburgh, Edinburgh, UK. [11]Colon Cancer Genetics Group, Institute of Genetics and Cancer, University of Edinburgh, Edinburgh, UK. [12]Centre for Clinical Brain Sciences, University of Edinburgh, The Queen's Medical Research Institute, Edinburgh BioQuarter, 47 Little France Crescent, Edinburgh, UK. [13]Edinburgh Cancer Research Centre, Institute of Genetics and Molecular Medicine, University of Edinburgh, Edinburgh, UK. [14]These authors contributed equally: Wei Xu, Ines Mesa-Eguiagaray. ✉e-mail: E.Theodoratou@ed.ac.uk; W.Cawthorn@ed.ac.uk

