## [Peer Review File · Nature Communications]

REVIEWER COMMENTS

Reviewer #1 (Remarks to the Author):

In this manuscript, Xu and colleagues have used data from UK Biobank to explore the genetic architecture underlying bone marrow fat fraction (BMFF). Using a deep learning algorithm, which they themselves developed and previously reported, on MRI scans from >45K biobank participants, they were able to measure the BMFF of the spine, femoral head, total hip, and femoral diaphysis. They then performed a GWAS in order to identify genetic signals influencing BMFF.

This is a novel and important resource that shines a light on a lesser known fat depot, and will be widely cited and used by the community. My one issue lies with the genetic analysis of the 6.4K non-white individuals in the study. At the moment, because of the small numbers, all the non-white participants with MRI data have been pooled together. I deeply appreciate the effort made to study 'non-white' genetics, but presented as it is, the data as it stands is uninterpretable. I think it would actually be more informative to present the data split into the different ethnicities. It will be underpowered of course. But it was going to be anyway, even if the 6.4K non-white individuals were all of the same ethnicity. Present the various frequencies of SNPs, and compare and contrast the ones that do or do not appear, fully acknowledging it is underpowered. I believe that this would be a more informative way of presenting the data.

Reviewer #2 (Remarks to the Author):

The authors have conducted a GWAS for bone marrow adiposity using data from the UK Biobank. Given the association of increased BMAT with osteoporosis and other diseases, and the unknown genetic determinants of BMAT, this study has significant scientific novelty. Bone marrow fat fraction (BMFF) was assessed using MRI imaging, with deep learning techniques measuring BMFF in the spine, femoral head, hip, and femoral diaphysis in >40K participants per site. The GWAS identified multiple genetic loci for BMFF at each site, many unique to specific regions. These loci were mapped to genes using nearest gene and cis-eQTL colocalization analyses. Sex-stratified GWAS revealed sex-specific effects, and cross-ancestry GWAS was performed for each bone region. Several biological pathways were implicated, particularly those related to adipose biology, bone density, and mesenchymal cell fate.

MAJOR COMMENTS

1. Ancestry definition: The stratification to "white" and "non-white" used in the paper is too simplistic and can obscure important genetic differences between ancestries. Please use more specific ancestry groups for stratifying the analyses and define how the ancestry was defined (based on self-reported ancestry, genetically determined ancestry, or their combination?).

2. The definition of independent SNPs, lead SNPs, and genomic risk loci does not follow a standard in the field. Defining independent SNPs by $r^2 < 0.6$ does not sufficiently separate independent signals. Even $r^2 < 0.1$ may not always distinguish truly independent signals, particularly if the LD patterns diverge from the reference population. Furthermore, observations in the UK Biobank suggest that $P < 5E-8$ may not be sufficient when a large number of SNPs are included in the analysis [PMID 28506277]. Please follow a more careful approach when describing these findings from the GWAS.

3. The authors have implemented all analyses with BMFF trait adjusted for BMI. Since BMI adjustment has a variable effect on genetic effect sizes depending on variant-specific pleiotropic associations, it would be informative to report the effect sizes with and without adjustment for BMI, at least for the lead variants.

4. Genetic correlation analyses are informative of the overall genetic relationship between traits. The authors report genetic correlation analyses between BMFF (BMI-adjusted) and BMD, BMI, and WHR. To complement these analyses, I recommend the authors perform further genetic correlation analyses with disease traits indicated by the GWAS associations (such as osteoarthritis, type 2 diabetes, coronary artery disease, breast cancer, stroke, blood pressure, and lymphocytic leukemia). Furthermore, since a genetic correlation using BMI-adjusted BMFF will not be informative of a possible relationship between BMI and BMFF, I recommend the authors also test the genetic correlation between BMI and BMFF unadjusted for BMI.

5. The authors show that sex and age influence BMFF on each site. It would be informative to also investigate sex and age differences in the genetic effects on BMFF. How many of the lead SNPs show significant interactions by sex and/or age?

MINOR COMMENTS

1. Abstract: The numbers in this sentence do not add up correctly: "... from MRI scans of over 45,000 participants in the UK Biobank imaging study, including >42,000 white and >6,400 non-white participants".

2. Where performing multi-ancestry GWAS, please clarify how many loci identified in the multi-ancestry GWAS are independent from those identified in white ancestry only.

3. Supplementary Tables 8-9, 26-27, and 39-40: Please consider adding effect allele frequency in the tables. This will be valuable for readers that wish to follow up on the findings and ensure they are looking at the correct allele.

REVIEWER COMMENTS

Reviewer #1 (Remarks to the Author):

In this manuscript, Xu and colleagues have used data from UK Biobank to explore the genetic architecture underlying bone marrow fat fraction (BMFF). Using a deep learning algorithm, which they themselves developed and previously reported, on MRI scans from >45K biobank participants, they were able to measure the BMFF of the spine, femoral head, total hip, and femoral diaphysis. They then performed a GWAS in order to identify genetic signals influencing BMFF.

This is a novel and important resource that shines a light on a lesser known fat depot, and will be widely cited and used by the community. My one issue lies with the genetic analysis of the 6.4K non-white individuals in the study. At the moment, because of the small numbers, all the non-white participants with MRI data have been pooled together. I deeply appreciate the effort made to study 'non-white' genetics, but presented as it is, the data as it stands is uninterpretable. I think it would actually be more informative to present the data split into the different ethnicities. It will be underpowered of course. But it was going to be anyway, even if the 6.4K non-white individuals were all of the same ethnicity. Present the various frequencies of SNPs, and compare and contrast the ones that do or do not appear, fully acknowledging it is underpowered. I believe that this would be a more informative way of presenting the data.

Response:

Thank you for your detailed review and valuable feedback on our manuscript. We have carefully considered your concern regarding the genetic analysis of the 6,400 non-white individuals in our study. As you pointed out, due to the small sample size and low power, we initially pooled all non-white participants together.

We agree with your suggestion that, despite the limited sample size, it would be more informative to conduct GWASes in the non-white sample by different ethnic groups. To address your concern, we have implemented the following changes (which we also explain in our response to Reviewer 2, Major Comment 1):

- We sub-grouped participants based on their ancestry as 'white' and 'Non-white' using UKBB data-field 22006 ("Genetic ethnic grouping"). 'Non-white' was further categorized into 'Asian', 'Black' and 'non-white mixed ethnic group' by data-field 21000 ("Ethnic background", as self-reported by participants). The sample size corresponding to each of these ethnicity categories are now shown in updated Supplementary Table 6, as shown below:

Supplementary Table 6. Summary characteristics of the non-white unrelated sample

	Femoral head (n=5933)	Total hip (n=6047)	Diaphysis (n=5844)	Spine (n=6367)
Age at imaging (years)*	63.89 (57.59-70.08)	64.00 (57.68-70.23)	63.77 (57.55-69.90)	63.89 (57.60-70.09)
Sex				
Male	2723 (45.84%)	2703 (44.64%)	2579 (44.13%)	2988 (46.93%)
Female	3217 (54.16%)	3352 (55.36%)	3265 (55.87%)	3379 (53.07%)

Ethnicity				
Asian	628 (10.58%)	650 (10.75%)	617 (10.56%)	664 (10.43%)
Black	270 (4.56%)	296 (4.89%)	285 (4.88%)	296 (4.65%)
Other non-white ethnic groups	5035 (84.86%)	5101 (84.36%)	4942 (84.56%)	5407 (84.92%)

*median, interquartile

- We conducted GWASes in ‘Asian’, ‘Black’ and ‘non-white mixed ethnic group’ for the non-white sample, and reported lead SNPs, independent significant SNPs, and mapped genes. The summary of non-white ethnic subgroups’ GWAS results are presented in Supplementary Tables 61-63.
- We performed power analyses of each GWAS in ‘Asian’, ‘Black’ and ‘non-white mixed ethnic group’, and discussed the low power in the limitations section.
- We updated our BMFF comparison between these ethnic groups for each bone region (Supplementary Figure 1; Supplementary Table 7).
- We have added a new study summary figure (main Figure 1) that highlights our multi-ancestry GWASes, including stratification into the Asian, Black, and non-white mixed-ethnicity participants, and the insufficient statistical power for GWASes in these groups.

The following specific changes have been made in the manuscript to address this comment:

Method (p23):

- *‘We sub-grouped participants based on their ancestry as ‘white’ and ‘Non-white’ using UKBB data-field 22006 (“Genetic ethnic grouping”). ‘Non-white’ was further categorized into ‘Asian’, ‘Black’ and ‘non-white mixed ethnic group’ by data-field 21000 (“Ethnic background”, as self-reported by participants).*
- *‘We performed GWAS power calculation using GCTA package (<https://yanglab.westlake.edu.cn/software/gcta/#GREMLpowercalculator>).*

Result (p12):

- *‘To further investigate genetic variants associated with different ethnic groups within the non-white population, we conducted GWAS analyses for Asian, Black, and mixed non-white ethnic groups (Supplementary Tables 61-63). We found three, two, and ten independent significant SNPs for ‘Asian’, ‘Black’ and ‘non-white mixed ethnic group’, respectively (Supplementary Table 62). Their beta estimate directions were consistent as in the white population. However, these non-white ethnic subgroup GWAS analyses were found to be underpowered (Fig. 1C; Supplementary Table 64).’*

Discussion (p21):

- *‘Stratification of the non-white participants into Asian, Black and mixed background showed that GWASes for these ethnicities were also underpowered (Fig. 1C; Supplementary Table 64). Furthermore, these results are prone to bias as a self-reported*

ethnicity parameter was used.'

Reviewer #2 (Remarks to the Author):

The authors have conducted a GWAS for bone marrow adiposity using data from the UK Biobank. Given the association of increased BMAT with osteoporosis and other diseases, and the unknown genetic determinants of BMAT, this study has significant scientific novelty. Bone marrow fat fraction (BMFF) was assessed using MRI imaging, with deep learning techniques measuring BMFF in the spine, femoral head, hip, and femoral diaphysis in >40K participants per site. The GWAS identified multiple genetic loci for BMFF at each site, many unique to specific regions. These loci were mapped to genes using nearest gene and cis-eQTL colocalization analyses. Sex-stratified GWAS revealed sex-specific effects, and cross-ancestry GWAS was performed for each bone region. Several biological pathways were implicated, particularly those related to adipose biology, bone density, and mesenchymal cell fate.

Response:

Thank you for your positive comments. We are pleased to hear that you recognize the novelty and significance of our study on the genetic determinants of bone marrow fat fraction (BMFF) using UK Biobank data. We are glad that you acknowledge our approach to assess BMFF using MRI imaging and deep learning techniques across various bone regions. Identifying multiple genetic loci for BMFF, including region-specific loci, and mapping these loci to genes through nearest gene and cis-eQTL colocalization analyses has indeed provided valuable insights. We also note your recognition of the sex-stratified and cross-ancestry GWAS efforts. The identification of several biological pathways related to adipose biology, bone density, and mesenchymal cell fate further highlights the broader implications of our findings. Thank you once again for your encouraging feedback.

MAJOR COMMENTS

1. Ancestry definition: The stratification to "white" and "non-white" used in the paper is too simplistic and can obscure important genetic differences between ancestries. Please use more specific ancestry groups for stratifying the analyses and define how the ancestry was defined (based on self-reported ancestry, genetically determined ancestry, or their combination?).

Response:

We thank reviewer for their insightful comments, which are echoed by those of Reviewer 1. We sub-grouped participants based on their ancestry as 'white' and 'Non-white' using UKBB data-field 22006 ("Genetic ethnic grouping"). Due to the small sample size (around n=6300, 13.5%), we initially pooled all non-white participants together. We agree with your suggestion that, despite the limited sample size (now shown in Supplementary Table 6), it would be more informative to conduct GWASes in the non-white participants by stratifying into more-precise ethnic groups.

To address this comment, we have made the following revisions:

- 'Non-white' has been further categorized into 'Asian', 'Black' and 'non-white mixed ethnic group' by data-field 21000 ("Ethnic background" as self-reported by UKBB participants). The sample size corresponding to each of these ethnicity categories are now shown in updated

Supplementary Table 6, as shown below:

Supplementary Table 6. Summary characteristics of the non-white unrelated sample

	Femoral head (n=5933)	Total hip (n=6047)	Diaphysis (n=5844)	Spine (n=6367)
Age at imaging (years)*	63.89 (57.59-70.08)	64.00 (57.68-70.23)	63.77 (57.55-69.90)	63.89 (57.60-70.09)
Sex				
Male	2723 (45.84%)	2703 (44.64%)	2579 (44.13%)	2988 (46.93%)
Female	3217 (54.16%)	3352 (55.36%)	3265 (55.87%)	3379 (53.07%)
Ethnicity				
Asian	628 (10.58%)	650 (10.75%)	617 (10.56%)	664 (10.43%)
Black	270 (4.56%)	296 (4.89%)	285 (4.88%)	296 (4.65%)
Other non-white ethnic groups	5035 (84.86%)	5101 (84.36%)	4942 (84.56%)	5407 (84.92%)

*median, interquartile

- We have conducted GWASes in ‘Asian’, ‘Black’ and ‘non-white mixed ethnic group’ for each bone region. The summary of non-white ethnic subgroups’ GWAS results are presented in Supplementary Tables 61-63.
- We have performed GWAS power calculations for non-white ethnic groups (method: <https://yanglab.westlake.edu.cn/software/gcta/#GREMLpowercalculator>) and the results are presented in Supplementary Table 64.
- We have cross-compared BMFF in white and non-white ethnic subgroups (Supplementary Figure 1; Supplementary Table 7).
- We have added a new study summary figure (main Figure 1) that highlights our multi-ancestry GWASes, including stratification into the Asian, Black, and non-white mixed-ethnicity participants.

Although we agree that the stratification to ‘white’ and ‘Non-white’ in the paper is simplistic, further stratification by ancestry in the non-white sample is likely prone to bias as it uses self-reported ethnicity. We found 3, 2, and 10 independent significant SNPs for ‘Asian’, ‘Black’ and ‘non-white mixed ethnic group’, respectively (Supplementary Table 62). Their beta estimate directions were consistent as in the white population. Power calculations showed that GWAS for ‘Asian’, ‘Black’ and ‘non-white mixed ethnic group’ were underpowered (Fig. 1C; Supplementary Table 64). [Results-Multi-ancestry meta-GWAS, p13]

We have acknowledged these limitations in Discussion, p21: *“Stratification of the non-white participants into Asian, Black and mixed background showed that GWASes for these ethnicities were also underpowered (Fig. 1C; Supplementary Table 64). Furthermore, these results are prone to bias as a self-reported ethnicity parameter was used.”*

Further details are provided in the updated Methods (p23) and Results (p12) of our revised manuscript.

2. The definition of independent SNPs, lead SNPs, and genomic risk loci does not follow a standard in the field. Defining independent SNPs by $r^2 < 0.6$ does not sufficiently separate independent signals. Even $r^2 < 0.1$ may not always distinguish truly independent signals, particularly if the LD patterns diverge from the reference population. Furthermore, observations in the UK Biobank suggest that $P < 5 \times 10^{-8}$ may not be sufficient when a large number of SNPs are included in the analysis [PMID 28506277]. Please follow a more careful approach when describing these findings from the GWAS.

Response:

We thank the reviewer for this comment. The approach we followed to describe our GWAS findings was based on established processes. In particular, we used FUMA to identify genome-wide significant SNPs ($P\text{-value} < 5 \times 10^{-8}$) in low LD ($r^2 < 0.6$) as independent significant SNPs. The P -value and r^2 thresholds we used are those suggested by FUMA: <https://www.ncbi.nlm.nih.gov/pmc/articles/PMC5705698/>

These thresholds were also used in many previously published GWASes (e.g., those listed below).

- <https://www.nature.com/articles/s41467-020-19378-5>
- <https://www.nature.com/articles/s41467-022-34732-5>
- <https://www.nature.com/articles/s41467-023-44079-0#Sec10>
- <https://www.ahajournals.org/doi/full/10.1161/JAHA.123.030661#d1e999>
- <https://www.ncbi.nlm.nih.gov/pmc/articles/PMC6836675/>
- <https://www.nature.com/articles/s41588-018-0151-7#Sec3>

The aim of our study was to identify candidate variants and loci for further fine-mapping and functional research. Given this, our primary goal was to (reasonably) minimize type II error, which may come at the cost of obtaining some false-positive results. With respect to the genome-wide significant P -value ($P\text{-value} < 5 \times 10^{-8}$), we argue that this is the threshold used in most GWASes, including those previously reported for UK Biobank. Therefore, it is logical for us to use the same thresholds so that we can compare our GWAS findings with those for other traits.

However, to address the reviewer's concern regarding a more-stringent threshold, we have made the following revisions:

- We defined genome-wide significant SNPs ($P\text{-value} < 5 \times 10^{-8}$) in low LD at an $r^2 < 0.3$ as independent significant SNPs and further subdivided these into independent lead SNPs if they are in approximate linkage disequilibrium ($r^2 < 0.1$). Using these more-stringent thresholds, the summary of lead SNPs and independent significant SNPs for the meta-GWAS in the white population (Supplementary Tables 19-20) and stratified by sex (Supplementary Tables 48-49) were updated. We found that there is substantial overlap of independent significant SNPs and lead SNPs identified using the more-stringent threshold ($r^2 < 0.3$) vs those identified using the less-stringent threshold ($r^2 < 0.6$). This overlap is presented in the upper portions of Supplementary Tables 23 (for white meta-GWAS) and 52 (for the sex-stratified meta-GWAS);

the relevant portion of Supplementary Table 23 is pasted below, for convenience. Please also see p24 of the manuscript.

Summary of lead and independent significant SNPs using two thresholds in meta-GWAS for the white population

P<5e-08	Femoral head			Total hip			Diaphysis			Spine		
	r ² <0.6	r ² <0.3	overlap	r ² <0.6	r ² <0.3	overlap	r ² <0.6	r ² <0.3	overlap	r ² <0.6	r ² <0.3	overlap
n.												
Lead SNPs	23	23	22	48	46	45	37	36	35	51	49	49
Independent significant SNPs	67	34	33	147	72	66	134	65	61	174	80	73

- We also defined genome-wide significant SNPs (P-value < 1 × 10⁻⁸) in low LD (r² < 0.3) as independent significant SNPs. The summary of lead SNPs and independent significant SNPs for meta-GWAS in the white population (Supplementary Tables 21-22) and stratified by sex (Supplementary Tables 50-51) have been updated. Comparisons of the independent significant SNPs and lead SNPs identified using the more-stringent threshold (r² < 0.3, P-value < 1 × 10⁻⁸) vs those using the less-stringent threshold (r² < 0.6, P-value < 5 × 10⁻⁸) are presented in Supplementary Tables 23 (for meta-GWAS in the white population) and 52 (for the sex-stratified meta-GWAS). For convenience, the relevant portion of Supplementary Table 23 is pasted below. Please also see p24 of the manuscript.

Summary of lead and independent significant SNPs using two thresholds in meta-GWAS for the white population

r ² <0.6; P<5e-08/ r ² <0.3; P<1e-08	Femoral head			Total hip			Diaphysis			Spine		
	r ² <0.6; p<5e-08	r ² <0.3; p<1e-08	overlap	r ² <0.6; p<5e-08	r ² <0.3; p<1e-08	overlap	r ² <0.6; p<5e-08	r ² <0.3; p<1e-08	overlap	r ² <0.6; p<5e-08	r ² <0.3; p<1e-08	overlap
n.												
Lead SNPs	23	17	16	48	38	38	37	27	27	51	36	36
Independent significant SNPs	67	25	25	147	58	54	134	52	51	174	62	57

Overall, there is a high degree of overlap of SNPs identified across different LD and P-value thresholds. While the more-stringent thresholds result in fewer SNPs, they do not substantially alter the main findings. This reinforces the reliability of our results obtained using LD r² < 0.6 and P-value < 5 × 10⁻⁸. We highlight this on p8 and p12 of our revised manuscript.

3. The authors have implemented all analyses with BMFF trait adjusted for BMI. Since BMI adjustment has a variable effect on genetic effect sizes depending on variant-specific pleiotropic associations, it would be informative to report the effect sizes with and without adjustment for BMI, at least for the lead variants.

Response:

We understand the concern regarding the impact of BMI adjustment on genetic effect sizes due to variant-specific pleiotropic associations. To address this, we have conducted GWAS analyses without BMI adjustment in the white population, both for the sexes combined (Supplementary Tables 14-18) and when stratified by sex (Supplementary Tables 43-47). We have not done this for the non-white populations, given that their small sample sizes are underpowered for GWASes.

Comparing the number of lead SNPs with and without BMI adjustment revealed that the same SNPs were identified for the head and spine analysis, and minimal differences were identified for total hip and diaphysis (1 SNP and 3 SNPs fewer without BMI adjustment, respectively). The direction of the effect estimates was consistent with and without BMI adjustment, with median beta differences around 1.6×10^{-5} (Supplementary Table 15).

Please find below a list of the changes made to the manuscript to address this comment:

- We have conducted the sensitivity analysis without BMI adjustment for meta-GWAS in the white population (Supplementary Tables 14, 16, 18) and stratified by sex (Supplementary Tables 43, 45, 47).
- We have added the comparison tables of meta-GWAS with/without BMI adjustment (Supplementary Tables 15, 17, 44, 46).
- We have described the methods (p23) and presented results of meta-GWAS without BMI adjustment in the white population in the manuscript (p8, p12).

4. Genetic correlation analyses are informative of the overall genetic relationship between traits. The authors report genetic correlation analyses between BMFF (BMI-adjusted) and BMD, BMI, and WHR. To complement these analyses, I recommend the authors perform further genetic correlation analyses with disease traits indicated by the GWAS associations (such as osteoarthritis, type 2 diabetes, coronary artery disease, breast cancer, stroke, blood pressure, and lymphocytic leukemia). Furthermore, since a genetic correlation using BMI-adjusted BMFF will not be informative of a possible relationship between BMI and BMFF, I recommend the authors also test the genetic correlation between BMI and BMFF unadjusted for BMI.

Response:

We thank the reviewer for this comment. To address this comment, we have made the following revisions:

- We have conducted cross-trait LDSC genetic correlation analyses between BMFF (BMI-adjusted), and the disease traits suggested (Supplementary Table 36). We have not done this for lymphocytic leukemia because no suitable GWAS statistics of lymphocytic leukemia can be used for cross-trait LDSC (https://www.ebi.ac.uk/gwas/efotraits/EFO_0000095).
- We have estimated the genetic correlation between BMI and BMFF for each of the bone regions without adjustment for BMI. In Supplemental Table 36 this is indicated with “§”.
- We have reported the cross-trait LDSC results in the manuscript (p10). We have updated the cross-trait LDSC method in the manuscript, including details of traits that can be used for cross-trait LDSC comparison (p25).

5. The authors show that sex and age influence BMFF on each site. It would be informative to also investigate sex and age differences in the genetic effects on BMFF. How many of the lead SNPs show significant interactions by sex and/or age?

Response:

We acknowledge the importance of understanding how sex and age may influence the genetic effects on BMFF.

To address this comment, we have made the following revisions:

- We have analyzed the Sex x Genotype interaction, using the formula to calculate t_{sex} and P_{sex} as developed by Winkler et al. [method detailed in Winkler et al., PLOS Genetics, <https://journals.plos.org/plosgenetics/article?id=10.1371/journal.pgen.1005378#sec017>]:

$$t_{sex} = \frac{b_M - b_F}{\sqrt{SE_M^2 + SE_F^2 - 2r_{sex} \times SE_M \times SE_F}}$$

This is described in the 'Methods' of our revised manuscript (p.23).

- We reported t_{sex} and P_{sex} for lead SNPs (Supplementary Table 53, p12 of the manuscript). We did not find P_{sex} genome wide significant for any of the lead SNPs in four bone regions.
- We understand that it is important to investigate Age x Genotype interaction effects. The age at imaging distribution of the UKBB multi-modal imaging participants in our study is 45-85 years old, with a median age of 65 and 98.26% of the participants being older than 50 years old. Given the age range of our study sample, this analysis is out of the scope of this paper.

MINOR COMMENTS

1. Abstract: The numbers in this sentence do not add up correctly: "... from MRI scans of over 45,000 participants in the UK Biobank imaging study, including >42,000 white and >6,400 non-white participants".

Response:

Many thanks for the comment. In the Abstract we have revised this to, "Herein, we used deep learning to measure the BMFF of the spine, femoral head, total hip, and femoral diaphysis from MRI scans of over 47,000 participants in the UK Biobank imaging study, including >41,000 white and >6,300 non-white participants."

2. Where performing multi-ancestry GWAS, please clarify how many loci identified in the multi-ancestry GWAS are independent from those identified in white ancestry only.

Response:

Many thanks for the comment. We have added one column 'found in meta-GWAS in white sample' in Supplementary Table 66, to specify whether the independent significant SNPs found in multi-ancestry meta-GWAS were also found in meta-GWAS in the white population. In summary, 46 (38.02%), 108 (34.39%), 65 (27.78%) and 94 (30.32%) significant SNPs in multi-ancestry meta-GWAS were found independent from those identified in the white population. We have added this summary in the main text of the sub-section 'Multi-ancestry meta-GWAS' (p13-14, manuscript).

3. Supplementary Tables 8-9, 26-27, and 39-40: Please consider adding effect allele frequency in the tables. This will be valuable for readers that wish to follow up on the findings and ensure they are looking at the correct allele.

Response:

Many thanks for the comment. We have added the effect allele frequency in these Supplementary Tables, which are now numbered 11-12 (previous tables 8-9), 39-40 (previous tables 26-27), and 65-66 (previous tables 39-40).

We thank both reviewers for their insightful suggestions, which we agree have improved the quality and scope of our revised manuscript.

REVIEWERS' COMMENTS

Reviewer #1 (Remarks to the Author):

The authors have satisfactorily responded to my concerns.